# Charge transfer as a ubiquitous mechanism in determining the negative charge at hydrophobic interfaces

Emiliano Poli[1✉], Kwang H. Jong[1] & Ali Hassanali[1✉]

The origin of the apparent negative charge at hydrophobic–water interfaces has fueled debates in the physical chemistry community for decades. The most common interpretation given to explain this observation is that negatively charged hydroxide ions (OH⁻) bind strongly to the interfaces. Using first principles calculations of extended air–water and oil–water interfaces, we unravel a mechanism that does not require the presence of OH⁻. Small amounts of charge transfer along hydrogen bonds and asymmetries in the hydrogen bond network due to topological defects can lead to the accumulation of negative surface charge at both interfaces. For water near oil, some spillage of electron density into the oil phase is also observed. The computed surface charge densities at both interfaces is approximately $-0.015 \, \mathrm{e/nm^2}$ in agreement with electrophoretic experiments. We also show, using an energy decomposition analysis, that the electronic origin of this phenomena is rooted in a collective polarization/charge transfer effect.

[1] International Centre for Theoretical Physics, Trieste, Italy. ✉email: epoli@ictp.it; ahassanali@ictp.it

Air bubbles and oil droplets in water move toward the anode under the presence of electric fields[1–3], implying that they develop a natural negative charge. This experimental observation has served as the seed for one of the most hotly debated topics in physical chemistry for the last several decades with numerous conflicting interpretations underlying the origins of the negative charge[4–8]. Here, we make an important leap in our understanding of this problem providing a framework that helps rationalize the observed phenomena that is rooted in what we propose is a generic charge transfer mechanism associated with the interfacial structure at the surface of water and oil. The electrokinetic experiments of both air and oil droplets paint a very similar picture despite having rather disparate chemistries namely, that the zeta potential ($\zeta$) vanishes under acidic conditions and that with increasing pH, it becomes increasingly negative showing that these interfaces retain a negative charge at neutral pH[3]. If waters constituent ions, the proton and hydroxide, are the only sources of charging in water, these experiments suggest that the negatively charged $OH^-$ ions stick to the surfaces with binding energies on the order of 10–20 times larger than thermal energy[5,6].

This interpretation has been heavily contested from both experimental and theoretical fronts. Spectroscopy of interfaces using second harmonic generation and sum frequency generation (SFG) tell a less consistent picture ranging from the presence of $H^+$ at the surface under acidic conditions to weak binding of the $OH^-$ under basic conditions[9–13]. There have also been other suggestions implicating hydrocarbon impurities such as bi-carbonate ions[14,15] as the source of the surface charge, although this seems to have been ruled out in very recent experimental work interpreting the Jones–Ray effect[16]. Theory and simulations continue to play an important role in the interpretation of these experiments. Several groups have pioneered insightful ab initio[17–19], empirical valence bond[20–22], and more recently, classical empirical potential based[23] molecular dynamics simulations of the air–water interface. Most of these studies indicate that the $H^+$ has some marginal preferential binding to the surface of water, whereas the $OH^-$ is effectively repelled from this interface.

In this report, we assert that the negative charge at extended hydrophobic interfaces does not require the binding of $OH^-$ ions. Using state-of-the-art linear scaling density functional theory (LS-DFT)-based simulations of thousands of atoms (see Fig. 1), we elucidate the electronic properties of two paradigmatic systems: the air and oil–water interfaces. Both systems are characterized by a regime of significant negative charge, that is primarily modulated by how charge transfer changes for different water defects[24]. At the surface of water, the charge transfer leads to a triple layer of charge with negative surface charge a couple of Angstroms from the surface (Fig. 2a). A similar effect occurs at the oil–water interface where additional complexity emerges: there is some transfer of charge from the water to the oil molecules, leaving the latter negatively charged (Fig. 2b). Interestingly, we show using energy decomposition analysis (EDA) that despite the low-dielectric character of the oil molecules, those at the surface experience subtle electronic effects involving both polarization and charge transfer. The surface charge densities that we determine are an order of magnitude larger than those discovered in previous studies[25–27] bringing them in closer agreement with experiments. Besides providing fundamentally new insights into the controversy, our results should have potential for broader impact. Interfaces such as the ones we have tackled here, also lie at the heart of fundamental questions in atmospheric[28], pre-biotic chemistry[29,30] in addition to electrophoretic experiments and triboelectrification[31]. The electronic and molecular origins of the underlying phenomena in these contexts, remains poorly understood. The charge transfer mechanisms espoused in this contribution will help broaden the scope of the discussion in the area of contact electrification.

## Results

### The air/oil water interfaces are negatively charged.
In the ensuing analysis, we begin by first discussing the charge gradients observed at the surface of water and at the oil–water interface. We subsequently compare and contrast the microscopic origins of the charge oscillations in the two systems.

The top left and right panels of Fig. 1 show snapshots of our two simulated systems: (a) the air–water and (b) the oil–water systems consisting of a total of 6540 and 13,480 atoms, respectively. The yellow-colored surface corresponds to the Willard–Chandler[32] instantaneous interface (WCI) that is constructed for the water phase. In both figures, the cyan-colored isosurface corresponds to the highest molecular orbital, which is localized at the interface. To the best of our knowledge, our simulations represent the first of their kind where the electronic structures of thousands of atoms of the air and oil–water interfaces are treated.

The possibility that a charge transfer mechanism could rationalize the negative charge at hydrophobic surfaces has been suggested in previous theoretical studies[25,26], although the reported charge densities were too small compared with those obtained from electrophoretic experiments. The essential idea is that asymmetries in hydrogen bonding between water molecules at an interface leads to a subsequent imbalance in the charge transfer along donating versus accepting hydrogen bonds. We begin by showing in Fig. 2 the charge densities and integrated surface obtained for the density-derived electrostatic and chemical (DDEC) charges extracted from our calculations for the air–water and oil–water interfaces. As alluded to in Fig. 1, in order to perform this analysis, a description of the corrugations of the interface is needed. We used a formulation proposed by Willard and Chandler[32], which characterizes the instantaneous density fluctuations of the interface. The zero on the $x$ axes of Fig. 2 corresponds to the position of the WCI.

The left panel of Fig. 2 reveals the presence of significant charge gradients at the air–water interface covering a length scale of ~5 Å. In particular, we observe a triple layer of charge (orange bars): from above the instantaneous interface up to 0.3 Å below it there is a positively charged layer (1st layer) of thickness ~2.0 Å with a charge density of ~0.22 e nm$^{-3}$; immediately below this, there is a compensating negative layer of ~1.5 Å thickness (2nd layer) but with a larger charge density of ~−0.41 e nm$^{-3}$; and finally, below 2–5 Å from the interface, there is another positively charged layer (3rd layer, charge density ~0.12 e nm$^{-3}$) after which, charge neutrality develops (the 4th layer). The presence of the triple layer is essentially caused by the asymmetry in the magnitude of the first two charged layers where the negative branch is about twice as large than that of its positive counterpart closer to the interface. The charge density shown can be integrated from the vacuum to the bulk to give a better description of the cumulative surface charge. The dashed-blue curve shows the integrated surface charge density—between 1 and 2 Å from the WCI interface, there is a substantial negative charge density of ~−0.015 e nm$^{-2}$, which is about an order of magnitude larger than previous findings[25,26].

How does the behavior change near the oil–water interface? Rather unexpectedly, the oil phase is not a passive spectator in the charging mechanism. The right panel of Fig. 2 shows similar distributions in charge density and integrated surface charge for both the water and the oil. As before, the zero on the $x$ axis corresponds to the position of the WCI surface. In stark contrast to the surface of water, there is a sharp peak of positive charge in the water phase just below the interface with a charge density of ~0.39 e nm$^{-3}$. This layer is followed by a negatively charge region

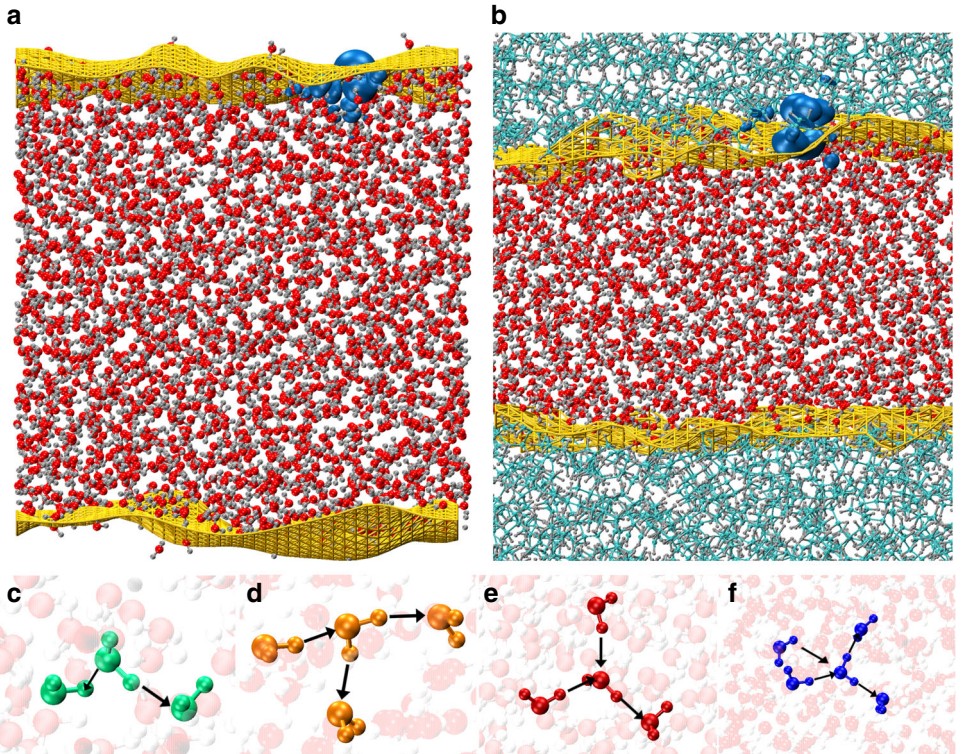

**Fig. 1 Simulated systems and coordination defects.** Outline of the two simulated systems: **a** the air–water and **b** the oil–water. The Willard–Chandler instantaneous surfaces are highlighted in yellow. The highest occupied molecular orbital (HOMO) in both systems is highlighted in cyan. **c**, **d**, **e**, **f** represent the different types of water molecules coordination configurations that will form the focus of our discussion later. For each coordination configuration the accepted hydrogen bond(s) are represented by an arrow pointing in, whereas the donated hydrogen bond(s) are shown by an arrow pointing out. For brevity, the following nomenclature will be used in the rest of the paper: donated hydrogen bonds will be called out, whereas accepted hydrogen bonds will be called in. In this way, **c**, **d**, **e**, **f** represent, respectively 1in-1out, 1in-2out, 2in-1out, and 2in-2out water coordination configurations.

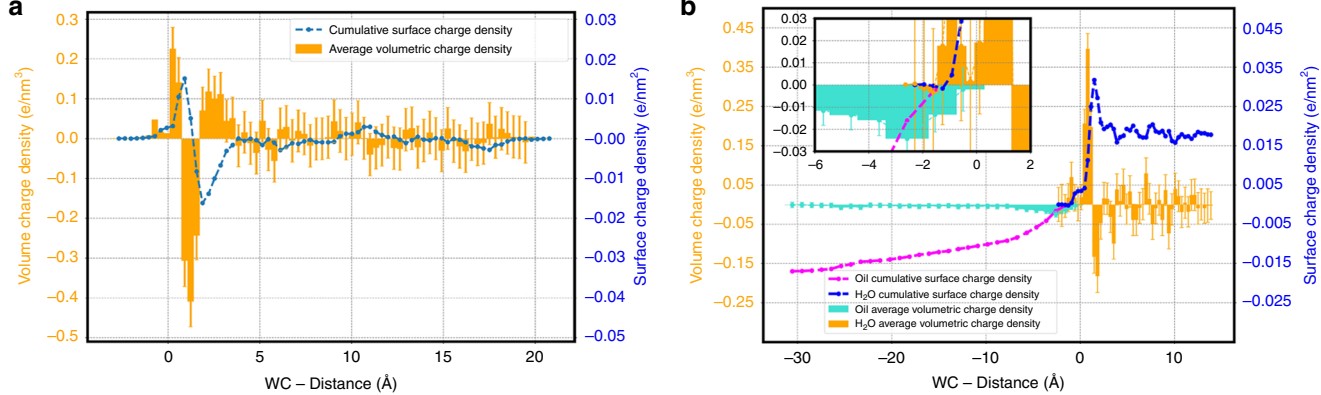

**Fig. 2 Volumetric and surface density profiles.** Volumetric charge density (orange bars) and integrated surface charge density (blue line) obtained for the DDEC charges extracted from our calculations for the air–water (**a**) and oil–water (**b**) interfaces. Different axis scales were used to fit both profiles in the same graph. The left axis refers to the volumetric charge density, whereas the right one refers to the integrated surface charge density. For the oil phase in the right panel, the charge density is reported using turquoise bars and the integrated surface charge density is shown by the magenta line. The inset in the right panel magnifies the charge oscillations right at the interface between the two phases. The error bars reprsent the standard deviations calculated from the single frames charge densities with respect to the average trend.

of water with a lower charge density of $\sim-0.18\,\mathrm{e\,nm}^{-3}$. The complementary charge distributions of the dodecane molecules forms one of our central findings in this report, namely that the oil phase is negatively charged. We observe a large negative surface charge density of $\sim-0.016\,\mathrm{e\,nm}^{-2}$ in the oil phase. These results are striking as they show that there is a net charge transfer of $\sim0.4$ electron charge from the water to the oil. Furthermore, the magnitude of this surface charge is very similar to what is

observed at the air–water interface, which is also consistent with the similarity in the zeta-potentials obtained from air and oil droplets[33].

**Charging is coupled to the local topology and environment.** In order to dissect the microscopic origins of the charge gradients observed at the interface, we turn next to examining how the

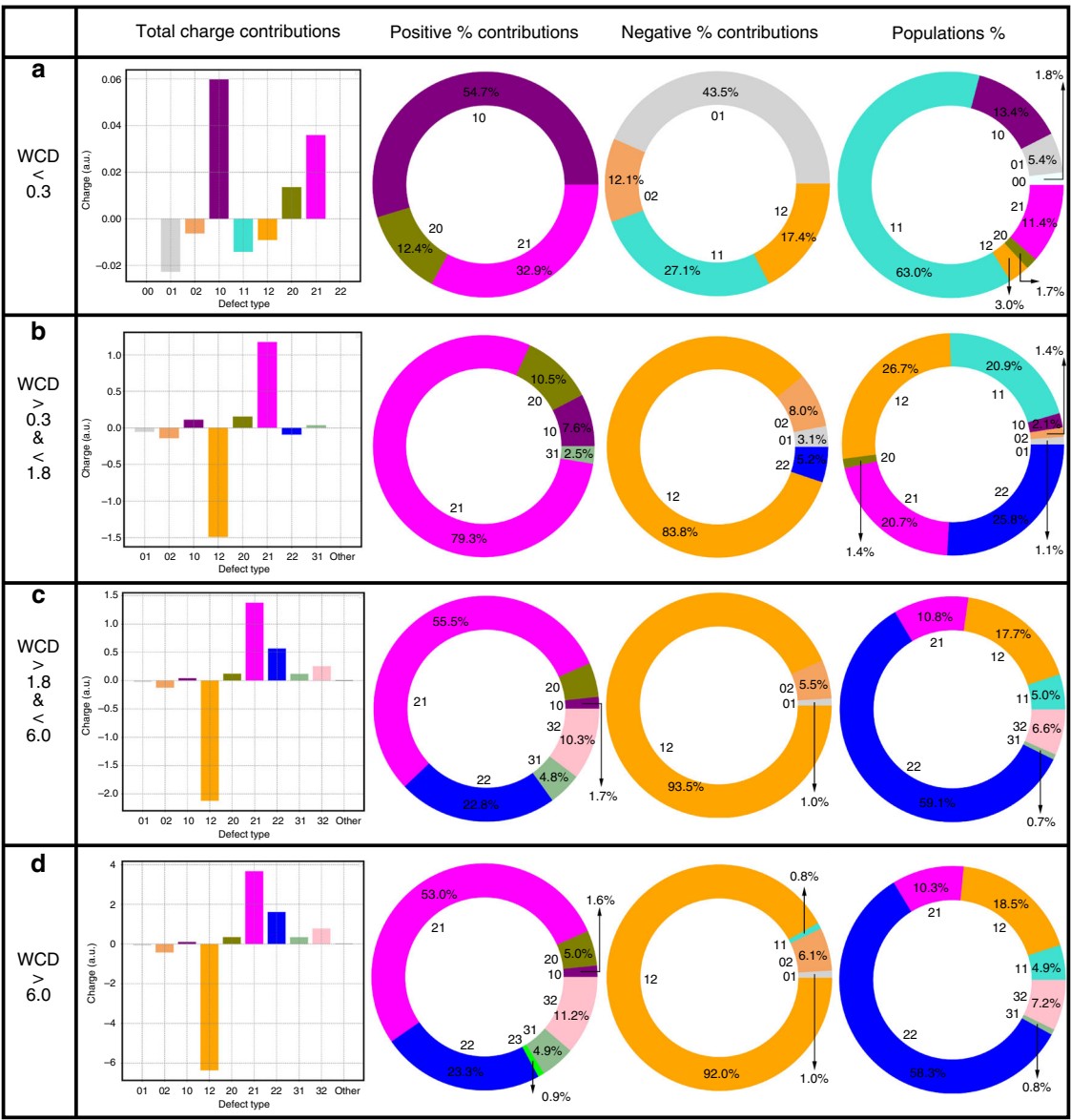

**Fig. 3 Water/air layer-by-layer charge and population contributions.** Analysis of the water–air interface layer-by-layer (as defined in the main text) in terms of the average total charge contribution, percentage contribution to the positive charge, percentage contribution to the negative charge and percentage contribution to the overall population for each water coordination configuration. **a** reports the different contributions for the first layer, **b** for the second layer, **c** for the third layer and **d** for the fourth one.

charge is modulated by differences in topological defects. It is well appreciated that fluctuations in liquid water both in the bulk[24] and at surface[34,35] create local coordination defects, which have asymmetries in the number of donated versus accepted hydrogen bonds. The bottom panel of Fig. 1 depicts different types of water molecules that will form the focus of our discussion later ranging from the canonical tetrahedral waters that accept and donate two hydrogen bonds (2in-2out) to various other undercoordinated defects such as those that accept one/two and donate two/one hydrogen bonds (1in-2out and 2in-1out waters as described in the caption).

Using the four layers previously defined to describe the different charged layers, we determined how the relative concentration of different water molecules change as one moves from the interface to the bulk, as well as their relative contribution to the total charge (3). For clarity, the contributions to the total charge are given separately for the species that are positive or negative. In all the layers, we show only the most dominant coordination states: for

the first layer, this involves all species with a total population >0.5%, whereas for the other layers it is those that contribute at least 2% of the total population.

The first layer (Fig. 3a) is dominated by many under-coordinated species owing to the presence of the dangling O–H bonds[36]. Particularly relevant is the role of the 1in-0out and 2in-1out species both of which lead to a net positive contribution of charge within the first layer. It is also striking to note that in spite of being balanced in terms of hydrogen bonds, the 1in-1out species noticeably contributes negatively to the overall charge underlining the importance that the asymmetry of hydrogen bonding itself does not exclusively control the charging behavior and that there are important collective polarization effects.

Layer two (Fig. 3b) presents some drastic differences with respect to first and it is this region that provides important clues into the origin of the negative charge at the surface of water. The negative oscillation in charge between 0.3 and 1.8 Å below the WCI is dominated by the competition between the charging

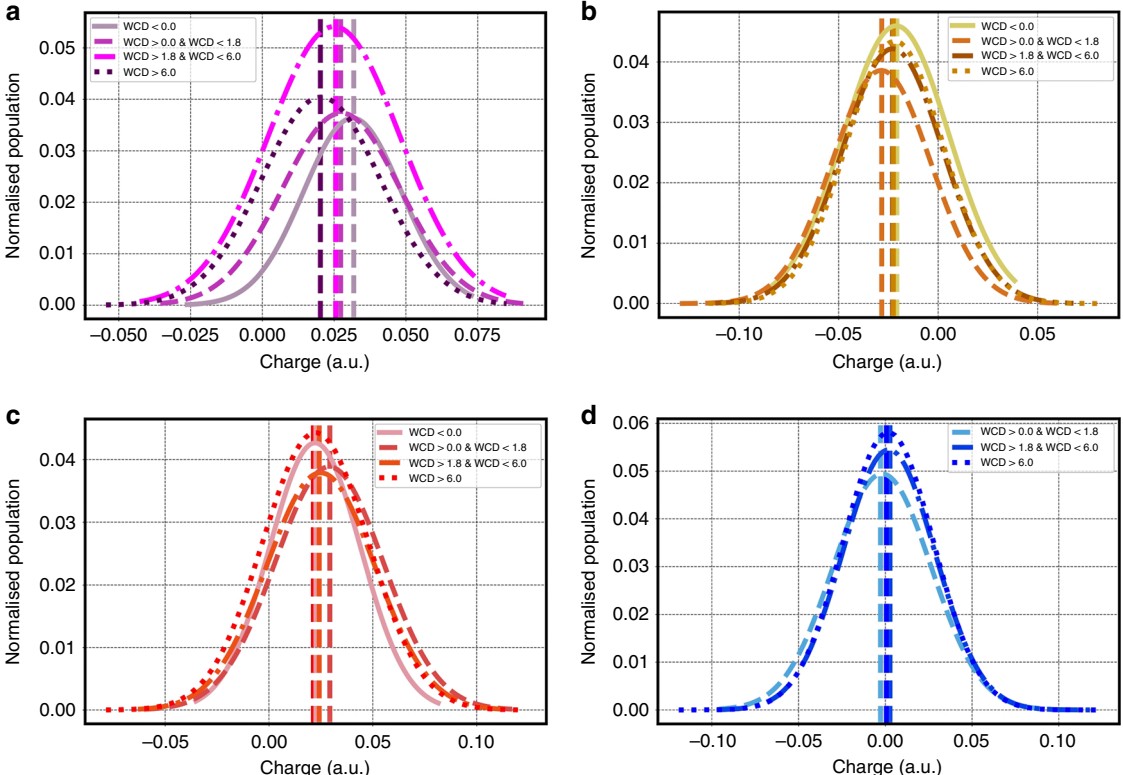

**Fig. 4 Water/air layer-by-layer charge distributions.** Layer-by-layer charge distributions of the 1in-0out (**a**), 1in-2out (**b**), 2in-1out (**c**), and 2in-2out (**d**) water molecules for the water–air interface. The first moment is highlighted by a vertical line for each distribution.

behavior of the 1in-2out and 2in-1out water molecules. Contrary to what one might have expected, the behavior is not symmetric. The total absolute charge of the 1in-2out waters is larger than 2in-1out by ~0.35 e. Interestingly, this behavior is not exclusive to the interface but is a feature that continues to occur even in the bulk. Part of this effect originates from the fact that the concentration of 1in-2out waters is larger than 2in-1out waters consistent with previous studies[24], although the role of this difference in the creation of charge gradients has not been recognized until this point. Besides this, we will see later that charging behavior of the these two types of water defects are not symmetric across the region of the interface.

Despite its relatively lower weight, it is interesting to underline the negative contribution that the tetrahedral waters give to the total charge in the second layer. This behavior then swings in the opposite direction in the 3rd (Fig. 3c) and 4th (Fig. 3d) layers where the 2in-2out waters integrate to a net small positive charge. Similar to the traits of the 1in-1out water molecules right at the top of the surface of water, we find that the charge fluctuations are affected by other factors beyond the asymmetry in hydrogen bonding. In addition, what is also particularly important to observe as we move across different layers is that the dominant positive–negative branches of the 1in-2out and 2in-1out waters occur even in the bulk phase. Charge neutrality in bulk liquid water involves a complex mix of the 2in-2out, 3in-1out, 3in-2out, and 2in-0out, essentially counterbalancing the negative contribution of the 1in-2out water molecules that is not achieved by the 2in-1out defects.

Figure 3 provides a collective picture of the role of water topology on charging but does not tell us anything about the nature of the fluctuations of individual molecules and how they are perturbed by the interface. In order to explicate this, we show in Fig. 4 the charge distributions of the 2in-1out, 1in-2out, 2in-2out, and 1in-0out water molecules in the four layers. These

distributions confirm our intuitions built on the preceding analysis that the interfacial region agitates water molecules in subtle and very surprising ways. There is clearly a change in the average charge for water molecules in different layers. A more quantitative analysis of the differences can also be obtained by the first four moments reported in Tables 1 to 4 of the Supporting Information. These results show that the for the water–air interface the average charge on each water molecule for the 1in-2out, 2in-1out species increase from the 1st to the 2nd layer (right at the negative oscillation) just to decay again in the bulk like region. An increment of the average charge for the 1in-0out molecules is also observed at the surface. This increment, however, decrease monotonically by moving into the bulk, affirming the important role that this species has mainly for surface properties. As previously reported, the tetrahedral coordinated waters in the 2nd layer are on average negatively charged and then regain a positive character toward the bulk. Although the average charge changes between the 2nd and 3rd layer are moderate (~0.005 e on average) the huge increment in weight of the 2in-2out species between these two layers lead to sizeable changes in the overall charge contributions. This observation further solidifies the assumption that small variation in the average charge per molecules can lead to big changes in the interfacial system behavior. These types of features would not be captured by models using a constant[25] or parameterized charge transfer[26] schemes that only partially respond to the local environment losing important collective polarization effects (more info can be found in Supplementary Fig. 8 of the SI).

A recent set of new experimental work using SFG of the surface of water demonstrated that the number of dangling O–H bonds near the interface, is ~25%[37]. How one defines the interface that is probed by the SFG experiment is non-trivial. However, using the definitions proposed by Bonn and co-workers, we computed the number of dangling bonds in the water model we used namely,

TIP4P/2005—the number of dangling O–H bonds we obtain is ~27 ± 2.4%, which is consistent with the experiments. Recall that the number of water molecules with dangling O–H bonds correspond to the topological defects that are 2in-1out or 1in-1out or 0in-1out (although these are found in very small concentration). Interestingly, Bonn and co-workers compare a whole class of empirical water models including mB-POL—the percentage of our dangling O–H bonds is consistent with these results. As we have also pointed out from our work, one of the key ingredients of observing the charge asymmetries is the apparent higher concentration of 1in-2out defects versus 2in-1out defects. This feature is not just exclusive to the interface but also in the bulk. In a previous study[35], we have compared, exhaustively, the concentration of defects in different water models ranging from classical empirical to DFT based and finally, of course mb-POL. Again, the conclusion is that the higher concentration of 1in-2out water molecules compared with 2in-1out is found in all water models including mb-POL. Having understood the origins of the charge oscillations at the surface of

water, we move next to examining how these properties behave near oil. In order to adequately compare the two interfaces, we repeated the charge and population analysis previously done for the water–air system as a function of different layers (Fig. 5) for the oil–water interface. The interfacial structure of water near the oil surface is quite different from that at the surface of water. The 1st layer (Fig. 5a) is characterized by an increased presence of the 2in-1out coordination defects. In addition, the 1in-1out water molecules contribute positively to the overall charge. These two effects conspire together to produce a much larger positive charge in the first layer compared with the surface of water. This is in essence activated by the transfer of charge to the oil phase. The 2nd layer (Fig. 5b) derives very similar trends with respect to its air–water equivalent, except for the fact that in this case, the tetrahedral 2in-2out water integrate to a net positive charge. Beyond the 2nd layer, the behavior is very similar to that observed in Fig. 3. Although the individual charge distributions for water near the oil show some differences (Fig. 6 and Supplementary Tables 1–4 SI), the overall behavior is very similar—specifically,

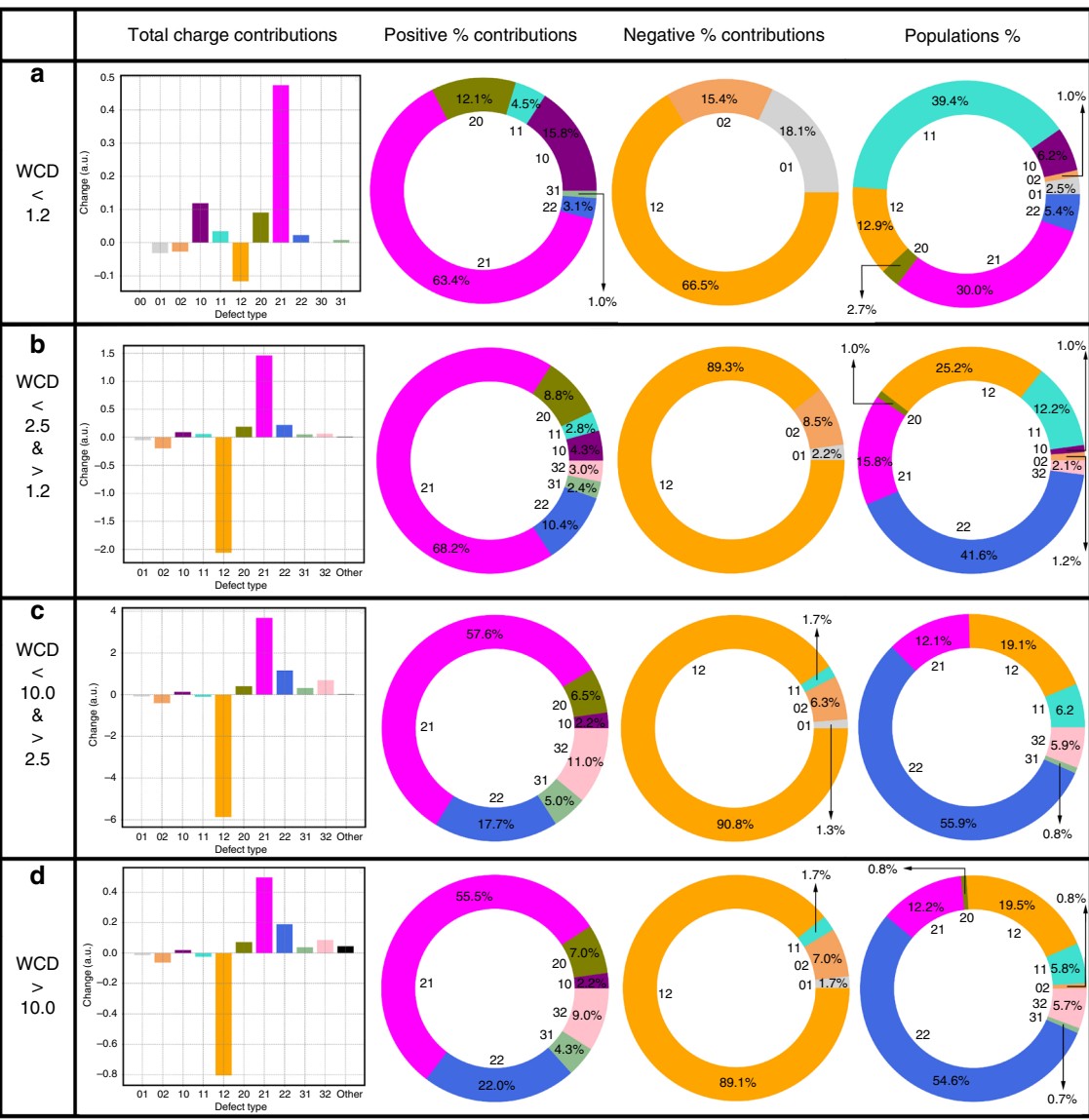

**Fig. 5 Water/oil layer-by-layer charge and population contributions.** Analysis of the water–oil interface layer-by-layer (as defined in the main text) in terms of the average total charge contribution, percentage contribution to the positive charge, percentage contribution to the negative charge and percentage contribution to the overall population for each water coordination configuration. **a** reports the different contributions for the 1st layer, **b** for the second layer, **c** for the third layer and **d** for the fourth one.

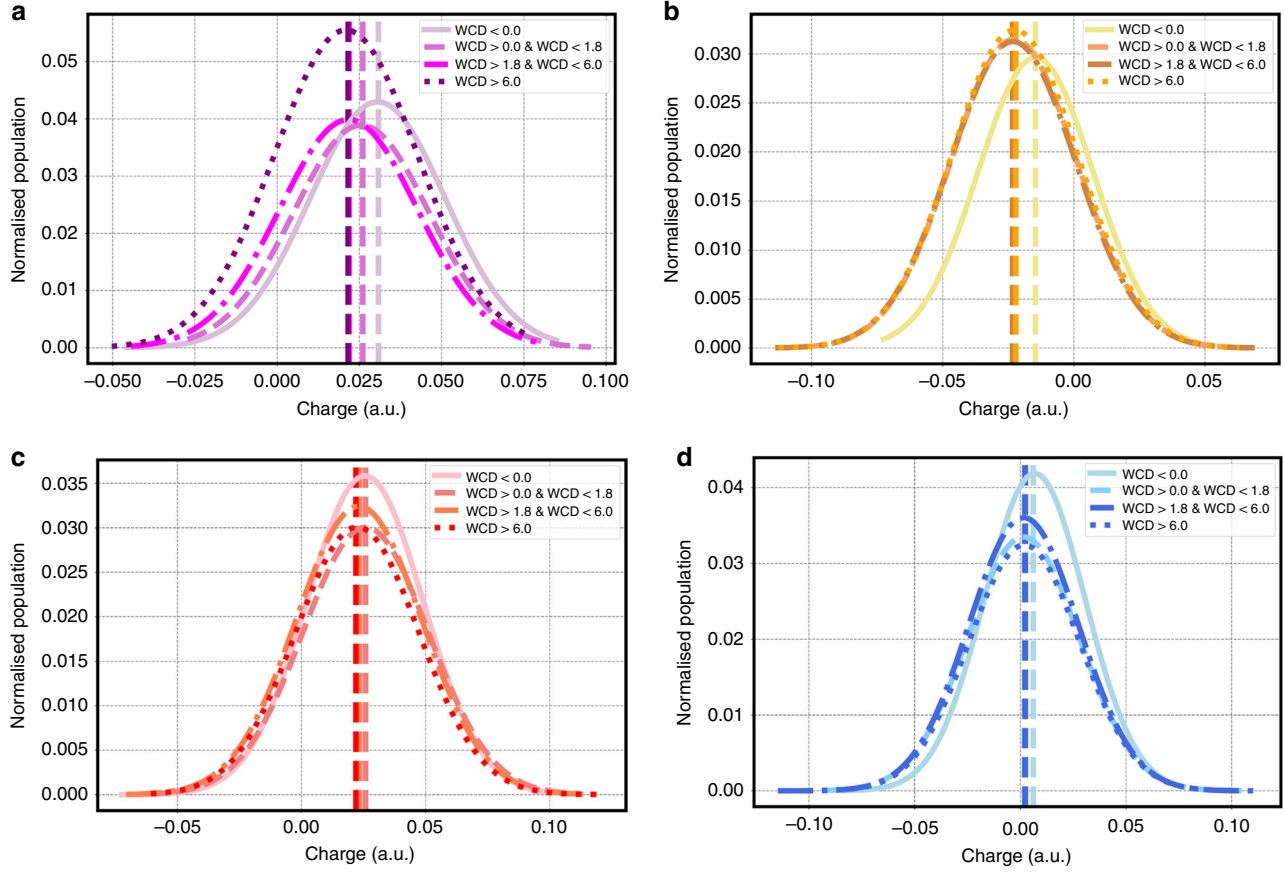

**Fig. 6 Water/oil layer-by-layer charge distributions.** Layer-by-layer charge distributions of the 1in-out (**a**), 1in-2out (**b**), 2in-1out (**c**), and 2in-2out (**d**) water molecules for the water–oil interface. The first moment is highlighted by a vertical line for each distribution.

charge fluctuations of different water topologies and their sensitivity to proximity to the interface appears to be an important part of the physics of charging.

Figures 4 and 6 for the air and oil–water interfaces show that the charge on different water molecules sustains rather large fluctuations. Furthermore, the fact that tetrahedral waters can conspire to produce a positive total charge suggests a highly non-trivial asymmetry in the charge transfer occurring on the donating versus accepting side of the hydrogen bond network. The origin of these asymmetries lies in the small instantaneous distortions of hydrogen bonds, which leads to fluctuations in the net charge of the tetrahedrally coordinated water molecule as seen in previous studies[38].

**The role of electronic polarization and charge transfer.** The stabilizing role of charge transfer between the hydrogen bonds of water molecules has been well appreciated in the literature[38,39]. Its role however, at the surface of water and how it is modulated by defect fluctuations has not been recognized until this point. The situation near the oil–water interface is much more surprising and warrants a deeper examination.

In order to understand better the underlying quantum mechanical effects associated with the build up of surface charge, we performed an EDA as implemented in ONETEP (Order-N Electronic Total Energy Package)[40]. The EDA analysis essentially provides a framework to disentangle various contributions of the interaction energy between the water and decane coming from electrostatics, polarization, exchange, and charge transfer. The EDA analysis reported here was performed on ~60 clusters consisting of one decane molecule and all water molecules within

3.5 Å from it. The clusters were carved out from the thermal simulations described earlier and typically consist of ~12 $H_2O$ molecules (see Fig. 7a). For this subset of clusters, the average DDEC charge on the decane molecules was −0.056 e.

We begin by first showing the qualitative behavior involving the intra and inter-fragment electron reorganization in the clusters. Fig. 7b, c shows the electron density difference (EDD) surfaces that are obtained from the EDA calculations between intermediate states involving the extraction of the polarization (blue isosurface) and charge transfer (yellow isosurface) contributions (see Methods for more details). Interestingly, we observe that the polarization EDD mostly involves reorganization along the backbone carbon atoms of the decane. On the other hand, charge transfer appears as response of the electron density that is mainly localized on the hydrogen atoms of decane. In both cases, nearby water molecules also exhibit perturbations from both effects.

A more quantitative measure of these effects at the oil–water interface can be explicated by examining the distributions associated with the various components of the interaction energy. Fig. 7d shows those components that are repulsive and attractive. The former are dominated by Pauli repulsion and exchange effects. On the other hand, charge transfer and polarization energetics contributes an attractive interaction that adds up to ~5 kcal mol$^{-1}$—for the clusters, this implies a binding energy of slightly under thermal energy per water molecule. As result of these charge transfer and polarization effects that render the oil-phase negative and the water one positive, the electrostatic component of the energy is also attractive (~5 kcal mol$^{-1}$). Our findings in the previous section strongly suggest that these effects

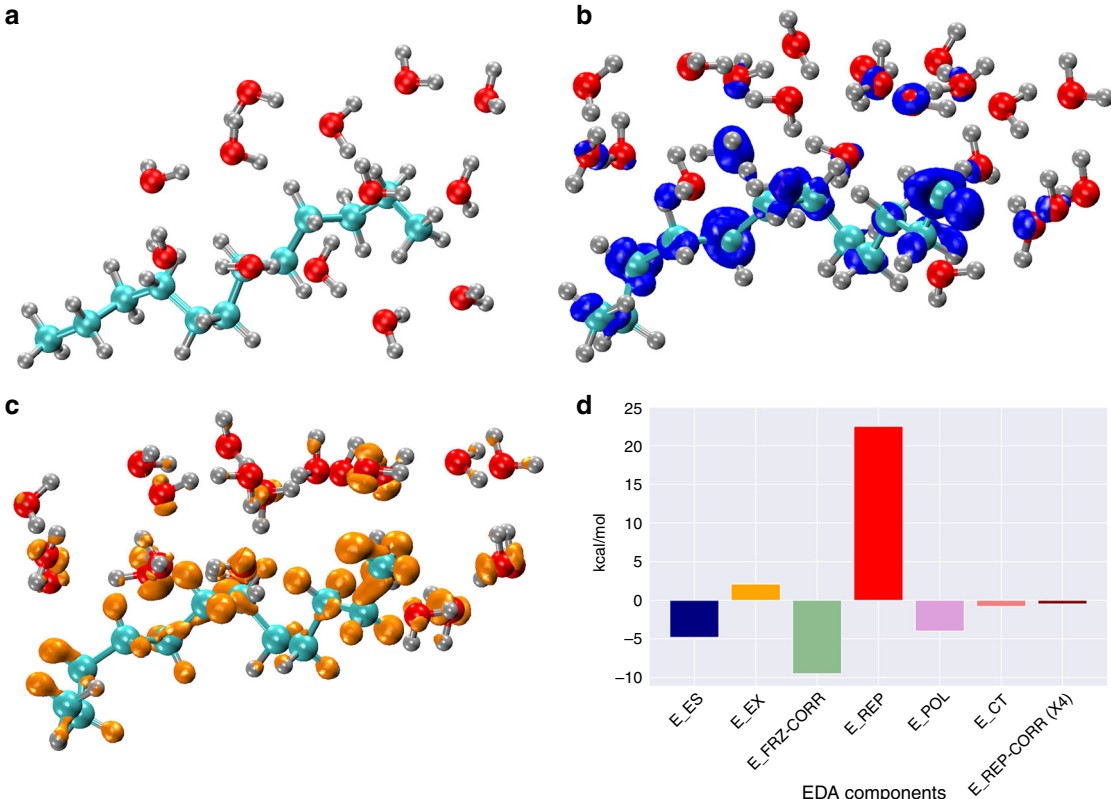

**Fig. 7 Water/oil clusters energy decomposition analysis.** Sample representation of the clusters considered for the EDA calculations **a**. The polarization (blue) and charge transfer (orange) electron density difference for the same sample oil–water cluster are highlighted, respectively, in **b** and **c**. The average energy component contributions calculated via EDA analysis for the 60 oil–water clusters are reported in **d**. The acronyms of the different components are: $E_{ES}$ is the Electrostatic Energy, $E_{EX}$ is the Frozen Exchange Energy, $E_{FRZ-CORR}$ is the Frozen Correlation, $E_{REP}$ is the Pauli Repulsion Energy, $E_{POL}$ is the (Stoll SCF-MI) Polarization Energy, $E_{CT}$ is the Charge Transfer Energy and $E_{REP-CORR}$ is the Repulsion Correlation.

are likely to be enhanced and also critical driving force in enhancing the negative surface charge observed at extended oil–water interfaces.

**Charge partitioning schemes and correlations**. The use of net atomic charges is widely used in the area of chemical sciences, as they provide a practical and convenient way to partition the electron density distribution into atomic or molecular contributions. The method of choice in this work was the DDEC charge partitioning. In the DDEC method, the net atomic charges are assigned as a functional of the electron density, hence are not sensitive to the choice of basis set. There are of course many charge schemes to partition the electron density with similar traits such as Bader[41] and the iterative Hirshfield (IH)[42] and stockholder (ISA)[43] approaches. The DDEC method combines the strengths of the ISA and IH methods with the additional trait of giving a more faithful representation of the potential $V(r)$. Although some of the criticism around the DDEC methods regards the inelegance of their formulation, their strength relies in an engineering approach that has been shown to lead to sound chemical results[44]. To assess the reliability of our results with respect to the charge partitioning adopted, we compared the results obtained with the DDEC scheme to the ones obtained using two different charge representations: natural bond orbital (NBO), iterative Hirshfiled (IH). These test were constructed by averaging over 20 and 10 frames for the air/water and oil/water interface, respectively, instead of the full 250 and 200 used in the original sets (owing to the computational cost). The results

depicted in Fig. 8 show that although there are subtle differences between the local behavior in the different charge representation, the main trends we observe on the charge transfer are consistent across all the three charge schemes.

In order to understand how the charge fluctuations are modulated by the local environment, we determined the correlation between the total charge on each $H_2O$ molecule (WATC) and various geometrical descriptors which are visually depicted in Fig. 9a–b. Figure 9c shows the Pearson correlation coefficient between the different geometrical parameters and WATC for the most populated water molecules in the different water layers relative to the interface. The four rows in the table correspond to the four regions relative to the WCD as shown in Fig. 3. For a more exhaustive analysis of the correlation matrices in all the layers, the reader is referred to the SI (Supplementary Figs. 9–12).

The proton transfer coordinate[45] ($\tau$) shows a strong negative correlation with respect to the central $H_2O$ WATC for those water molecules accepting a HB from it (i.e., $A_\tau 1$, $A_\tau 2$). A strong positive correlation is instead observed for waters molecules donating a HB to the central $H_2O$ (i.e., $D_\tau 1$, $D_\tau 2$). These results show that the more the proton is shared, namely a smaller $\tau$ between the molecules accepting a HB and the central $H_2O$, the higher is the charge transferred to it making the central water more negatively charged. For the case of $D_\tau 1$ and $D_\tau 2$, the proton transfer coordinate has a similar role but now the charge is transferred from the central water to the other $H_2O$ molecules making it more positive. For this reason, we observe a positive correlation in this case.

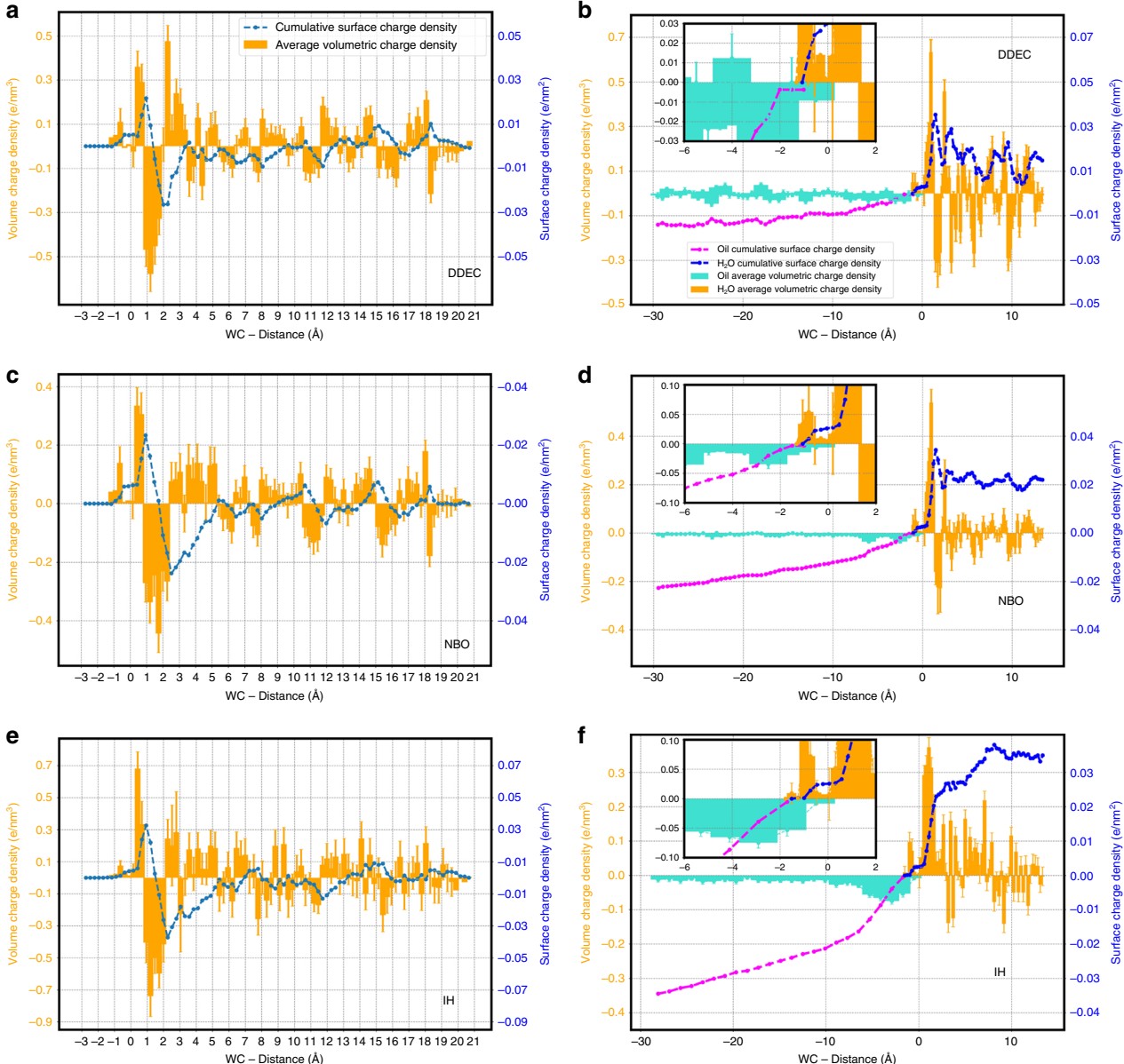

**Fig. 8 Charge scheme validation.** Comparison of the charge profiles obtained for 20 test frames using IH, NBO, and DDEC charge schemes for both water/air (**a**, **c** and **e**) and water/oil (**b**, **d** and **f**) interfaces. The same colors and axis scales from Fig. 2 were used here. The error bars represnt the standard deviations calculated from the single frames charge densities with respect to the average trend.

Although the charge transfer is much less correlated with the angular descriptors, the effects are not insignificant. We observe a positive correlation between WATC and the angle defined by the central water molecule, $A_\tau 1$ and $A_\tau 2$. Conversely, a negative correlation is observed for the angle between the central $H_2O$, $D_\tau 1$, $D_\tau 2$. These relations seem to suggest that the bigger is the angle formed by the central water molecule and the $H_2Os$ accepting a HB from it, the more positive is WATC. This effect could be ascribed to a more-effective alignment of the bond dipoles that results in a larger charge transfer from the central $H_2O$ to its hydrogen bond partners. This is also reflected in the correlation between the charge of the central water and AA1 angle described in the caption of Fig. 9. Clearly, the fact that the correlations we report along these geometrical parameters represent only a subset of the reaction coordinates involved in modulating the charge transfer, and warrants further investigation.

## Discussion

The findings of our work give strong indication that different types of charge transfer mechanisms can lead to a significant build up of surface charge density at hydrophobic interfaces. One of the essential ingredients associated with this phenomena is the presence of local topological defects involving undercoordinated water molecules at the interface. These types of waters are most susceptible to asymmetries in the magnitude of charge transfer between the accepting and donating side, which can leave them with a slight bias to take on either a positive or negative charge. The large surface charges we observe compared with previous studies[25,26] arises from a combination of several different factors such as the higher concentration of 1in-2out versus 2in-1out water molecules, the larger average charge associated with each water species and finally, the asymmetries associated with the fluctuations in the charge of these water molecules. It is also clear from the analysis of the charge and correlations with geometrical

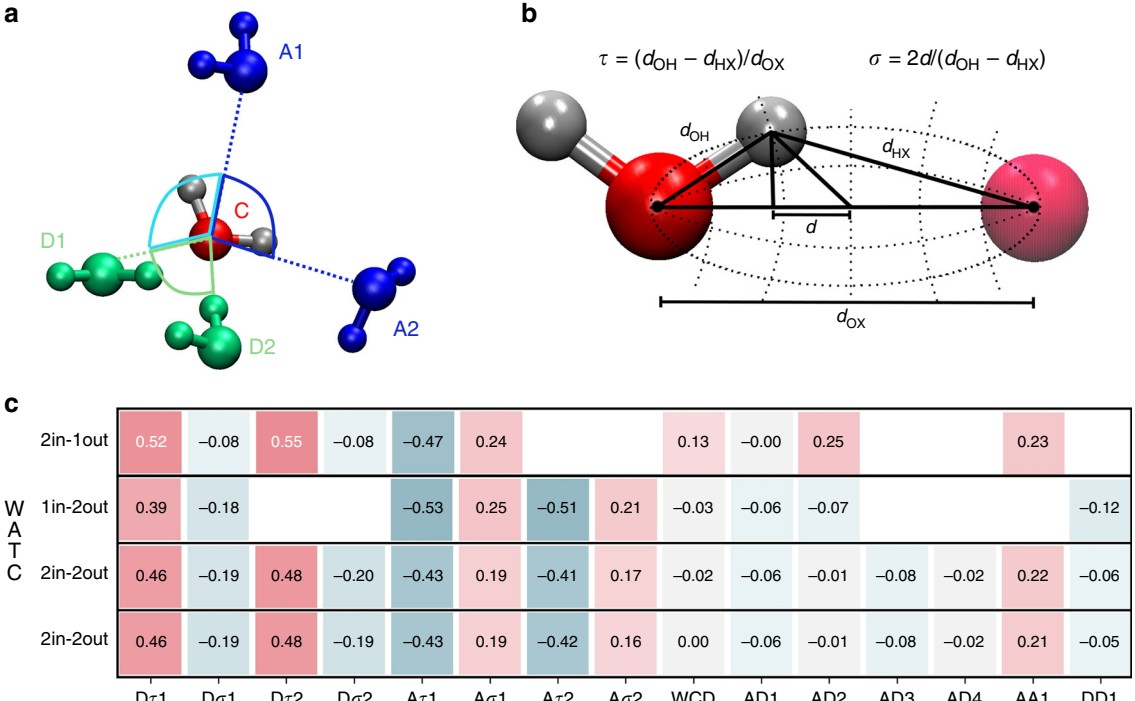

**Fig. 9 Correlation between water charge and geometrical descriptors. a** Angular coordinates for a tetrahedral coordinated water as reported in ref. [45]. The A tag is referred to the $H_2O$ molecules accepting a hydrogen bond from the central water. The D tag is referred to the $H_2O$ molecules donating an hydrogen bond to the central water. The subscript for each tag refers to the distance from the central water (i.e., $A_1$ is the nearest water to the central one accepting an OH bond from it). **b** Illustration of the proton transfer coordinates (PTC) in elliptical coordinates reproduced from ref. [77]. **c** Most relevant correlation coefficients (Pearson) calculated between the charge of the central water molecule (WATC) and the features shown in **a** and **b**. Only the results for the 2in-1out $H_2O$ in the 1st layer (1st row), 1in-2out in the 2nd layer (2nd row), 2in-2out in the 3rd and 4th layers (3rd and 4th row, respectively) are reported for brevity. The full correlation matrices for each species in every layer are reported in the Supplementary Information. The shorthanded labels for the PTC are: $D_\tau x$ and $D_\sigma x$ that indicate the quantities shown in **b** for D1 and D2, respectively. The same notation is applied for A1 and A2 (i.e., $A_\tau x$, $A_\sigma x$). WCD is the distance of the water molecule with respect to the Willard–Chandler interface. For the angular features, AA1 represents the $\widehat{A1CA2}$ angle, DD1 the $\widehat{D1CD2}$ angle, AD1 the $\widehat{A1CD1}$ angle, AD2 the $\widehat{A1CD2}$ angle, AD3 the $\widehat{A2CD1}$ angle, and AD4 is the $\widehat{A1CD1}$ angle.

parameters that the charge transfer is a rather complex process involving the coupling between several different degrees of freedom.

Near the surface of oil, something quite surprising happens: water defects can now inject some electron density into the oil phase, leaving the latter negatively charged. The effective surface charge densities derived at the surface of oil are very similar to that near the air–water interface and is also in very good agreement with measurements from electrophoretic experiments. Lurking behind the link between the simulations and these experiments should be placed in Fig. 2.

Roke and co-workers proposed a qualitative mechanism (see Fig. 5 of ref. [26]) for the effect of charge transfer on electrophoretic mobility. In this scenario, the slip plane resides several Angstroms from the oil surface and the role of fluctuations of water topologies is not considered. For the oil–water interface, our findings suggest that the slip plane will be pinned right near the oil droplet, as it is negatively charged, whereas the water in close proximity is positively charged. On the other hand for the air bubbles, the slip plane must reside somewhere in between the boundary of the second and third charged layers (shown in the left panel of Fig. 2).

A set of experiments that may also deserve another look at are surface sensitive photoelectron spectroscopy. Winter and Jung-wirth combined both theory and experiments in a series of papers to show that the valence band edge of sodium hydroxide solution could be used to interrogate the presence of hydroxide ions at the

surface of water[46]. They found that the presence of the hydroxide ion could be identified by an enhancement in the states close to the valence band. Although it is beyond the scope of the current report, we examined the projected density of states on water molecules residing in the negatively charged layer at the surface of water and found that they contribute substantially to the valence band (see Supplementary Fig. 14). Qualitatively, the defects at the surface are ~0.5 eV higher in energy than the mostly tetrahedral ones in the bulk (see Supplementary Fig. 14 upper left panel)— this indicates an uncanny similarity between the anionic defect, the hydroxide ion, and defects in neutral water.

Although it is beyond the scope of the current contribution to make quantitative predictions of spectroscopic measurements that could be performed to validate our observed results, we would like to suggest possible experimental routes that could corroborate our findings. In particular, non-linear vibrational spectroscopy techniques such as three-pulse photon echo (TPPE) spectroscopy experiments of bulk liquid water, indicate the presence of short-time oscillatory behavior on the sub-picosecond timescale[47]. In this direction, Kuhne and co-workers have very recently demonstrated that HB strength and HB charge transfer can be used as metric to predict HB rearrangements dynamics. These metrics have been used to represent spectral observables of non-linear spectroscopy experiments and have been proved to be in good agreement with correlation functions derived from TPPE experiments[47]. In addition, Tanimura and co-workers have recently demonstrated that in order to reproduce the Raman spectra in the Tera-Herz regime of bulk liquid water, a model that

accounts for intra and intra-molecular charge transfer needed to be included[48]. Accounting for these quantities lead, in fact, to a closer representation of the experimental results in bulk water. Using these type of spectroscopies on interfacial systems is still a challenging problem but nonetheless, in the future, these techniques could be used to study hydrophobic interfaces.

The mechanism associated with the response of the electronic degrees of freedom, as revealed by an EDA, involve a combination of both polarization and charge transfer. The coupling between the electronic reorganization and nuclear coordinates, such as local topology, opens up some very interesting perspectives and questions for future lines of research. An obvious one centers around the generality of our results to other extended hydrophobic interfaces or more heterogeneous surfaces such as metal oxides or even biological systems. In addition, the patchiness of surface charge at different interfaces deserves further investigation[49]. Our current models do not account for the possible binding of protons or hydroxide ions to the interface. Inserting the hydronium and hydroxide ion into the calculations and examining how it affects the charge transfer mechanisms will be the subject of a forthcoming study. At this point, we can only make speculations about what the presence of a proton would do to the charge at the air–water interface. Assuming that the proton exists as a local ionic defect, namely on a single water molecule as an Eigen or on two water molecules as a Zundel, we do not expect it to change the concentration of hydrogen bond topologies significantly. However, the presence of the ions will likely also change the extent of charge transfer between the water molecules at the interface as observed in previous studies[50,51]. The details of the charge transfer in the presence of the proton will also depend a lot on whether it is pinned above the Willard–Chandler interface or if it lies below it. If the proton lies within the second layer, we estimate that one would need a single proton per ∼10 nm$_2$ to neutralize the negative charge caused by the topological defects in that layer. Of course, another important ingredient would be the role of counterions. Previous studies have shown that while charge transfer does not affect the binding affinity of ions to the surface of water, ions can have long-range effects on surface charge[51]. More specific answers to these questions will the subject of a forthcoming study.

## Methods

**Classical molecular dynamics simulations.** Most of our results rely on simulations of two different systems that serve as prototypical models for water near hydrophobic surfaces: the surface of water and water near oil. The air–water interface was modeled building a water slab with dimension 40 Å × 40 Å × 40 Å and adding a vacuum padding of 80 Å on each side of the slab along the $z$ direction. This system comprises of a total of 6540 atoms (2180 water molecules). The second system studied a water–oil interface where the oil phase was composed of 200 dodecane molecules, whereas the water phase was made up of 1960 water molecules, leading to a total of 13,479 atoms. The cell dimensions for this system are 46.269 Å × 46.269 Å × 62.768 Å Our strategy for performing the analysis involved two steps: first, classical empirical molecular dynamics simulations were performed in order to allow for large and long timescale fluctuations that would not be possible using ab initio molecular dynamics; second, configurations from these simulations were sampled from which the electronic structure calculations were performed. The classical MD simulations were run using the GROMACS software[52]. The water phase was modeled using the TIP4P/2005[53] force field in both cases. The dodecane molecules were simulated using the modified OPLS-AA (L-OPLS) potential developed by Böckmann et al.[54]. This potential has been parameterized on the basis of high level ab initio calculations, densities, and heats of vaporization of both short- and long-chain alkanes and on the phase transition temperature of pentadecane in order to extend the OPLS-AA validity to long hydrocarbon chains and recover a more precise description of their phase transition temperatures and ordering. The air–water system was equilibrated for 10 ns using the NPT ensemble using the Parrinello-Rahaman barostat[55] for the first half of the run for bulk water, followed thereafter by an NVT simulation at 300 K, opening up a gap in the $z$ direction, which separated the two water surfaces by 160 Å for the remainder of the simulation. The production simulations were run for 20 ns. The surface tension computed from the classical simulations is 68.2 mN m$^{-1}$ in agreement with previous studies[56]. From these simulations a total of 250

configurations were randomly selected to perform the electronic structure calculations. The oil–water interfaces were equilibrated first 20 ns via NVT simulations. The production calculations were then run for 40 ns using the isothermal–isobaric ensemble. In all, 200 frames were randomly selected for the electronic structure calculations.

**Linear scaling density functional theory calculations.** In order to extend the scope of our electronic structure calculations, we employed a LS-DFT approach as implemented in the ONETEP code[57]. This technique allowed us to extend the system sizes in our study to thousands of atoms and to model electronic effects of extended hydrophobic interfaces on the nanometer length scale. For a more detailed and technical summary of the underlying theory the reader is referred to relevant literature[58]. Here, we briefly summarize the essential ideas. LS-DFT as implemented in ONETEP makes use of the nearsightedness[59] inherent to quantum many-body systems by exploiting the single-particle density matrix, $\rho(\mathbf{r}, \mathbf{r}')$[60,61] representation of the system of interest. Within ONETEP, $\rho(\mathbf{r}, \mathbf{r}')$, is expressed in a separable form[62,63] via atom-centered functions (non-orthogonal generalized Wannier functions, NWGFs[64]), $\phi_\alpha(\mathbf{r})$, as:

$$\rho(\mathbf{r}, \mathbf{r}') = \sum_{\alpha\beta} \phi_\alpha(\mathbf{r}) \mathbf{K}_{\alpha\beta} \phi_\beta^*(\mathbf{r}') \tag{1}$$

In the above, $\mathbf{K}_{\alpha\beta}$ are the matrix elements of the density kernel, which are nonzero only if $|\mathbf{r}_\alpha - \mathbf{r}_\beta| < \mathbf{r}_c$, with $\mathbf{r}_\alpha$ and $\mathbf{r}_\beta$ representing the coordinates of the centers of $\phi_\alpha$ and $\phi_\beta$, and $\mathbf{r}_c$ is a real-space cutoff threshold. The truncation of the density kernel ($\mathbf{K}_{\alpha\beta}$) is validated by the exponential decay of $\rho(\mathbf{r}, \mathbf{r}')$ with respect to $|\mathbf{r} - \mathbf{r}'|$ for systems with an electronic band gap[65]. Such truncation leads to a sparse density matrix ($\rho(\mathbf{r}, \mathbf{r}')$) that makes any insulating or semi-conducting systems (including the different interfaces considered here) treatable using linear scaling simulation. The non-orthogonal generalized Wannier functions (NGWFs) are centered on the nuclear coordinates and localized within a sphere of radius $r_\alpha$. Their non-orthogonality, implies a non-diagonal overlap matrix, $\mathbf{S}_{\alpha\beta}$:

$$\mathbf{S}_{\alpha\beta} = \int d\mathbf{r} \phi_\alpha^*(\mathbf{r}) \phi_\beta(\mathbf{r}) \tag{2}$$

In practice, the NGWFs are expressed as a linear combination of coefficients $C_{m\alpha}$, of localized but orthogonal periodic cardinal sine (psinc) functions[64], $D_m(\mathbf{r})$, as:

$$\phi_\alpha = \sum_m C_{m\alpha} D_m(\mathbf{r} - \mathbf{r}_m) \tag{3}$$

with $m$ indexing the real-space Cartesian grid inside the spherical localization region of $\phi_\alpha$. The psinc functions are obtained from a discrete sum of plane-waves, that makes the set of $D_m(\mathbf{r})$ independent of the nuclear coordinates and systematically improvable upon increase of the kinetic energy cutoff[64]. The convergence of the ONETEP approach is then dependent on interlinked computational factors such as the kinetic energy cutoff, the number of NGWFs ($\phi_\alpha$) per atom and their localization radius. In our LS-DFT calculations, the adopted kinetic energy cutoff was 1000 eV and four NGWFs were used for O atoms and 1 NGWF was used for the H atoms. In all cases, no truncation of the density kernel ($\mathbf{K}_{\alpha\beta}$) was enforced. The localization radius for the NGWFs was 10 Bohr in all cases. These parameters were chosen after a careful benchmark of the water monomer and dimer properties against the ab initio code CP2K[66]. Simulations were performed using the BLYP[67,68] functional with Grimme's D2[69] empirical dispersion corrections. In all cases, separable (Kleinman–Bylander)[70] norm-conserving pseudopotentials constructed with the Opium code[71] were used.

**DDEC charge analysis.** In order to characterize any possible charge gradients developing at our simulated interfaces, atomic point charges are then derived from the electron density. The atomic charges reported in this work were calculated using the DDEC3[72] scheme implementation in ONETEP[73]. DDEC3 is an atom in molecules approach where the total QM electron density ($n(r)$) is partitioned into overlapping atomic densities ($n_i(r)$):

$$n_i(r) = \frac{w_i(r)}{\sum_k w_k(r)} n(r) \tag{4}$$

The atomic partial charges are then computed by integrating the atomic electron densities over all space:

$$q_i = z_i - N_i = Z_i - \sum n_i(r) d^3r \tag{5}$$

where $N_i$ is the number of electrons assigned to atom $i$ and $z_i$ is its effective nuclear charge. In the same fashion, higher-order atomic multipoles may be computed as first-order, second-order, (etc) moments of the atomic electron densities. Various definitions of the weighting factors $w_i(r)$ exist. In the DDEC case, the weighting function is described so that the atomic weights are simultaneously optimized to resemble the spherical average of $n_i(r)$ and the density of a reference ion of the same element with the same atomic population $N_i$. In this way, the assigned atomic densities yield a rapidly converging multipole expansion of the QM electrostatic potential and the computed populations are chemically reasonable. A more detailed description of the method can be found in the following references[72,73]. An

important part of our findings reported in this work is the charge transfer occurring between different types of water molecules at the surface and between water and dodecane at the oil–water interface. The sensitivity of our results with respect to the choice of the functional, charge scheme, and sampling configurations was assessed and validated. In particular, we tested the charge scheme used, comparing the DDEC charges to the ones obtained via NBO, IH analysis. Regarding the functional, we validated the data obtained using BLYP+D2 against the ones obtained using the Van de Waals functional VV10[74] (and for few frames B3LYP[75]). In addition, we compared the results recovered using BLYP+D2 on smaller clusters with the ones obtained considering both hybrid functionals such as B3LYP as well as wavefunction approaches such as MP2[76]. The use of smaller clusters was forced by the heavy computational cost that such methods require. Besides the quality of the electronic structure, we also examined the sensitivity of the charge transfer to sampling configurations of the air–water interface sampled from mb-POL. The mb-POL potential reproduces many structural and dynamical properties of water across the phase diagram, and although we could not simulate a system as big as the one with TIP4P/2005, this test provide a useful comparison with respect to our results. More details about these benchmark simulations are expanded upon in the Supporting Information.

## Data availability

The data that support the plots within this paper are available from the corresponding author upon reasonable request.

## Code availability

Our results were obtained using a commercially available Linear Scaling DFT code named ONETEP. LS-DFT and EDA calculation inputs are available from the corresponding authors upon reasonable request.

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

## Acknowledgements

E.P. and A.H. acknowledge the support from KAUST (Shaheen II project-k1263). E.P. and A.H. acknowledge Himashu Mishra for the useful discussions and insightful help provided during the work.

## Author contributions

E.P. performed the classical molecular dynamics on the water/air slab and the LS-DFT and EDA calculations. K.H.J. performed the classical molecular dynamics on water/oil slab. E.P. and A.H. performed the data analysis and wrote the paper and created the figures. All the authors discussed the results and commented on the manuscript.

## Competing interests

The authors declare no competing interests.
