## [Peer Review File · Nature Communications]

Reviewers' comments:

Reviewer #1 (Remarks to the Author):

Understanding hydrophobicity is key to a myriad of problems in chemistry, physics and biology. The generality and importance of the topic is beyond doubt. The suggestion that subtle charge transfer effects are key to understand and characterize hydrophobic effects, is an important idea in my opinion, very much supported by the simulations presented in the paper. I believe the authors are presenting ideas on hydrophobicity that, if confirmed by experiments, would change drastically the way we look at hydrophobic interactions in materials science and biology. I thoroughly enjoyed the way the paper is written and the ideas put forward with clarity and careful calculations.

I suggest to consider the following revisions:

I would de-emphasize the discussion of 'numbers', and I suggest to just focus on trends and main qualitatively findings.

I would present **detailed** suggestions for experiments on how to identify the effect discussed in the paper and specific measurements to be made and how the specific measurements would in turn be corroborated by simulations, once carried out (even if not all spectroscopic calculations can be done at present).

Reviewer #2 (Remarks to the Author):

The authors present a computational analysis of the air-water and air-oil interfaces that attempts to explain the observation of a net negative charge in the water phase at the interface. This question has been a source of contention in the physical chemistry community for several decades. In this work, the authors suggest, based on their analysis of the electronic structure, that features such as charge transfer along hydrogen bonds and asymmetries in the water hydrogen bond network can be responsible for accumulation of negative charge in some of the layers close to the interface. I found this explanation an intriguing one that could make an important contribution to the ongoing debate. However, as the water structure near the interface and the charge partitioning approach are essential to the conclusions, some questions relating to these issues need to be addressed by the authors before the manuscript can be reconsidered for publication in Nature Comm.

Realizing that the system sizes employed by the authors are rather large, the use of a classical force field to generate an ensemble of structures on which to perform the electronic structure calculations seems reasonable, and a comparison with the high-level mB-POL force field was made for a small ensemble of configurations generated using this force field on a smaller system size. However, what seems to be missing is an attempt to make the sampled structures from either force field consistent with the chosen DFT scheme, something that could affect the results. It would seem that what one would want to do is to relax the configurations generated from a classical force field to a local minimum of the DFT model. Alternatively, the selected configurations could be subject to some local ab initio molecular dynamics generated by the DFT model. Either of these schemes should yield structures for further analysis that are more consistent with the DFT scheme. Can the authors comment on why they did not feel that this was necessary? A more convincing case could be made if

they could show that by carrying out this test for some of their configurations, the results are not affected.

Related to this point, the authors should comment on how the hydrogen-bond topologies they obtain from their force field calculations compare with previous ab initio molecular dynamics or empirical valence-bond model studies of the air-water or air-oil interface. For example, I believe that in the original study of Kuo and Mundy [Science 303, 658 (2004)] (surprisingly not cited by the authors), it was found that (in the nomenclature of the authors of the present manuscript) 1-in/1-out configurations at the surface layer are important, yet the authors only mention 1-in/0-out and 2-in/1-out topologies. Thus, I think it is important for the authors to address topologies that arise in other studies, compare with theirs, and discuss why they think they have hit on the right topologies. After all, this would seem to be a strong influencing factor in their charge analysis and could affect the results significantly. If possible, comparison with experimentally determined (and possible computationally determined) SFG spectra would also be useful. Are the findings of the authors consistent with what these spectra suggest?

Concerning the charge partitioning scheme, the analysis presented in the main manuscript is based entirely on the DDEC scheme. The SI shows some comparisons between DDEC and NBO for different functionals, but there is really no discussion about why DDEC was chosen and why a comparison between DDEC and NBO should be enough to convince the reader that the scheme was fully vetted. There are numerous charge-partitioning methods available, including old standards such as RESP, and more modern approaches such as additive variational Hirshfeld (from the work of Heidar-Zadeh and Ayers), to name a few. In order to avoid the impression that the authors cherry-picked the method (which I'm sure they did not) to support a particular picture, I think what would make a more convincing case would be to present results in the main manuscript that employ several charge partitioning schemes and demonstrate that the overall conclusion about charge transfer among the interfacial layers is not dependent on this choice (even if the results are not entirely the same quantitatively). That is, one would like to see some robustness of this key conclusion rather than having to have faith that DDEC is really producing the "right" picture.

Finally, one could levy the criticism that using neutral water is too ideal and not entirely realistic. If we believe previous ab initio MD results suggesting a preference of hydronium ions for the interface, then in a real interfacial system, there might be a small population of hydronium ions at the surface, the possibility of which is neglected here. This is not to say that the authors should repeat their study with a hydronium ion, but it would be interesting for them to estimate, based on their results, how the presence of an H_3O^+ ion at the surface (which, admittedly, given their system size, would be a high population) would affect their charge partitioning and hydrogen-bond topologies in the vicinity of the ion and how this might affect their integrated charge densities. Although probably a small perturbation, the results could be sensitive to the presence of this ion.

There are also a few small technical points the authors should address:

- i. Grimme's D2 scheme is not the more accurate of the empirical van-der Waals corrections. It was shown, for example, by Ma and Tuckerman to overestimate the density of bulk water [see J. Chem. Phys. 137, 044506 (2012)]. Why opt for D2 when D3 is superior and, I believe, available in CP2K? Perhaps, for single-point calculations in the water phase, the difference might not be large, but for the oil phase, I would expect a much large effect. Perhaps this could be addressed in the SI.

ii. On page 5, the surface tension value of 68.2 needs units.

Reviewer #3 (Remarks to the Author):

This manuscript focuses on how charge redistributes near an interface, as a result of partial charge transfer (CT) between molecules. This is done for both an oil/water and a vapor/water interface. To successfully address these interfaces requires large systems and they are able to perform density functional theory (DFT), using about 2000 particles. They use conventional molecular dynamics (MD) to generate configurations. A notable achievement of this work is the DFT calculations on such large systems. They are able to do this using linear scaling DFT. The focus of these calculations is to determine if CT creates a charged interface, as indicated by previous work using potential models (references 24 and 25). This may provide an explanation for electrokinetic properties of air and oil bubbles in water, which appear to have a negative charge. These calculations do find a charge layering—from the top of the interface towards the bulk, first a small positively charged layer, then a larger negatively charged layer, then another positively charged layer, then the neutral bulk. This layering, and the interpretation of that, as coming from hydrogen bond imbalances, is similar to references 24 and 25, except the present results are an order of magnitude larger, bringing them close to what might be required for the electrokinetic mobilities. They find similar results for the both the vapor/water and oil/water interfaces, except in the oil case, the oil acquires a negative charge.

Overall, this is an important, very well carried out and well written paper. I have a few suggestions to make this a stronger paper.

1. On page 10, it is mentioned that the charge transfer mechanism has been proposed in references 7, 24 and 25, but my reading of reference 7 makes no mention of charge transfer..so the 2012 papers (references 24 and 25) were the first to really make this suggestion, with quantitative estimates of charge transfer. Although in 1985 Collins and Washabaugh (Quarterly Review of Biophysics) made the suggestion that charge transfer leads to a negatively charged water surface. (They got some of the details wrong about the direction of charge transfer and the types of hydrogen bonds lost at the surface though). Ref. 7 is solely concerned with rejecting the hydroxide hypothesis.

2. I feel that the reasons the surface charges are so much bigger than the previous calculations are not fully explained. From Figure 8 of the supplementary information (I would suggest that figures 8 through 11 are actually tables) it looks like there is a charge transfer of 0.02 or 0.03 e for each hydrogen bond, which is the same as used in references 24 and 25. (A molecule acquires a charge of -0.02 if it donates a hydrogen bond and gets a charge of +0.02 if it accepts a hydrogen bond) . Molecules with even numbers of donated and accepted hydrogen bonds are roughly neutral (as given on Figure 8 of the SI and figure 3 of the main paper). So why is the charge so much greater than the earlier studies? The authors make the point that even the water molecules with equal number of donated and accepted hydrogen bonds ("2in-2out" or "2 2") have a net charge, but that contribution is not all that great. This is definitely true in the important negatively charged layer (from 0.3 to 1.8 Angstroms above the interface). Here is a largely the difference between negatively charged 1 in - 2out (1 2) and the positively charged 2 in - 1 out (2 1) and there are just more 1 2 than

2 -1, so that layer is negatively charged. Is the case that TIP4P/2005 (the model used to generate the configurations) has much more hydrogen bond imbalances than the SPC/E or charge transfer models used in the previous studies? Is it the use of the Willard-Chandler instantaneous surface, that effectively eliminates surface roughness so that layers are more obvious? (Vacha, et al, J. Phys. Chem. Lett. vol 3, p 107, 2012 found something like this) Or is there some other effect, not captured by charge transfer through hydrogen bonds?

3. On page 15, it is mentioned that some features may not be captured by models which use constant charge transfer, not modulated by the environment. Note that the charge transfer models of refs 24 and 25 do have a charge transfer that changes with distance, so not constant, and responsive to the local environment, at least in terms of hydrogen bond lengths. It would be interesting to see what other environmental effects might be relevant, other than counting hydrogen bonds. For example, using a tetrahedral order parameter or a Voronoi volume analysis. Does the charge depend on molecule's free volume for instance? Also, note the authors label the 2 in 2 out molecules as "tetrahedral" but having that hydrogen bond structure does not necessarily mean a tetrahedral local geometry, so better to avoid that term unless an analysis does show they are tetrahedral.

4. On the bottom of page 14, they refer to Figures 7 through 11 of the SI, but they must mean figures 8 through 11 (and they should really be tables, as suggested above).

With some further clarification about why the charges are so much bigger than previous estimates this would be a stronger paper. Note that the magnitude is important if it is the explanation for the electrokinetic results. Otherwise, this is an interesting, important and well-done paper.

Rebuttal Nature Communication

Emiliano Poli,* Kwang H. Jong, and Ali Hassanali*

*Condensed Matter Statistical Physics Department, The Abdus Salam International Center
for Theoretical Physics, Strada Costiera, Trieste*

E-mail: epoli@ictp.it; ahassanali@ictp.it

Reviewer # 1 :

Understanding hydrophobicity is key to a myriad of problems in chemistry, physics and biology. The generality and importance of the topic is beyond doubt. The suggestion that subtle charge transfer effects are key to understand and characterize hydrophobic effects, is an important idea in my opinion, very much supported by the simulations presented in the paper. I believe the authors are presenting ideas on hydrophobicity that, if confirmed by experiments, would change drastically the way we look at hydrophobic interactions in materials science and biology. I thoroughly enjoyed the way the paper is written and the ideas put forward with clarity and careful calculations.

I suggest to consider the following revisions: I would de-emphasize the discussion of 'numbers', and I suggest to just focus on trends and main qualitatively findings. I would present *detailed* suggestions for experiments on how to identify the effect discussed in the paper and specific measurements to be made and how the specific measurements would in turn be corroborated by simulations, once carried out (even if not all spectroscopic calculations can be done at present).

We thank the reviewer the positive comments and for acknowledging the broader implications of our results on hydrophobicity. In fact, we do believe that if these results can be validated experimentally, it will change the framework through which we all think about hydrophobic interactions in both the material science and biological physics. If our manuscript is accepted, our results will motivate new experiments to interrogate the observations we make.

The reviewer makes the following two suggestions:

1) De-emphasizing the discussion of the “numbers”, and I suggest to just focus on trends and main qualitative findings.

2) Making a stronger connection to suggestive experiments that can be done to validate the proposed mechanisms.

1) The first suggestion is rather challenging for us to address since the other two reviewers have made recommendations to be more quantitative about the numbers we report. It is also important to keep in mind that one of the experiments we are intending to compare with in these simulations are electrophoretic measurements suggesting negative zeta potentials of air-bubbles or oil-droplets in water. In addition, our goal is to revisit existing charge-transfer models that have been used to rationalize this effect as a source of the negative charge and hence we believe it is important to compare and contrast our numbers compared to those in the literature.

2) We thank the reviewer for the second suggestion. We have now significantly expanded our discussion in the “discussion/conclusion” of the paper where we talk about various spec-

troscopic experiments that may provide a way probe the charge-transfer phenomena. These include: X-Ray Photoelectron Spectroscopy, Three-Pulse Photon Echo Spectroscopy and finally TeraHerz Raman Spectroscopy. In addition, we also discuss some older electrochemistry experiments that showed the importance of charge mobility along alkyl-chains in liquid alkanethiol monolayers on mercury electrode. The charge transfer that we see from the water to the oil-phase is also consistent with these observations. We have now added the following text to the paper after the paragraph where had originally already discussed the possible connection with X-ray photoelectron spectroscopy.

Although it is beyond the scope of the current contribution to make quantitative predictions of spectroscopic measurements that could be performed to validate our observed results, we would like to suggest possible experimental routes that could corroborate our findings. In particular, non-linear vibrational spectroscopy techniques such as three-pulse photon echo (TPPE) spectroscopy experiments of bulk liquid water, indicate the presence of short-time oscillatory behavior on the sub-picosecond timescale.¹ In this direction Kuhne and co-workers have very recently demonstrated that HB strength and HB charge transfer can be used as metric to predict HB rearrangements dynamics. These metrics have been used to represent spectral observables of non-linear spectroscopy experiments and have been proved to be in good agreement with correlation functions derived from TPPE experiments.¹ In addition Tanimura and co-workers have recently demonstrated that in order to reproduce the Raman spectra in the Tera-Hertz regime of bulk-liquid water, a model that accounts for intra and intra-molecular charge transfer needed to be included.² Accounting for these quantity lead in fact to a closer representation of the experimental results on bulk water. Using these type of spectroscopies on interfacial systems is still a challenging predicament nonetheless in the future these techniques could be used to study hydrophobic interfaces.

Reviewer # 2 :

The authors present a computational analysis of the air-water and air-oil interfaces that attempts to explain the observation of a net negative charge in the water phase at the interface. This question has been a source of contention in the physical chemistry community for several decades. In this work, the authors suggest, based on their analysis of the electronic structure, that features such as charge transfer along hydrogen bonds and asymmetries in the water hydrogen bond network can be responsible for accumulation of negative charge in some of the layers close to the interface. I found this explanation an intriguing one that could make an important contribution to the ongoing debate. However, as the water structure near the interface and the charge partitioning approach are essential to the conclusions, some questions relating to these issues need to be addressed by the authors before the manuscript can be reconsidered for publication in Nature Comm.

1) Realizing that the system sizes employed by the authors are rather large, the use of a classical force field to generate an ensemble of structures on which to perform the electronic structure calculations seems reasonable, and a comparison with the high-level mB-POL force field was made for a small ensemble of configurations generated using this force field on a smaller system size. However, what seems to be missing is an attempt to make the sampled structures from either force field consistent with the chosen DFT scheme, something that could affect the results. It would seem that what one would want to do is to relax the configurations generated from a classical force field to a local minimum of the DFT model. Alternatively, the selected configurations could be subject to some local ab initio molecular dynamics generated by the DFT model. Either of these schemes should yield structures for further analysis that are more consistent with the DFT scheme. Can the authors comment on why they did not feel that this was necessary? A more convincing case could be made if they could show that by carrying out this test for some of their configurations, the results are not affected.

We thank the reviewer for these important observations and questions. Our choice of the protocol we adopted, stems from a combination of technical/practical issues as well as important advancements that have been made in the understanding of theoretical models used to simulate bulk liquid water over the last decade. Below, we highlight and therefore justify our strategy. One of the important findings of our result is the how the charge transfer along hydrogen bonds couples with local coordination/topological defects in the hydrogen bond network. There are two important questions one pose here: firstly, how sensitive is the charge transfer to the quality of the electronic structure used (see discussion later on this topic) and secondly, how sensitive is the charge transfer to the sampling protocol, namely, the underlying nuclear positions or configurations which are eventually used to perform the electronic structure calculations. Given that we wanted to use large scale Linear-Scaling Density Functional Theory (LS-DFT) calculations to study large extended hydrophobic-water interfaces, performing *ab-initio* molecular dynamics simulations is computationally prohibitive at this point. On the other hand, it is not clear at all that sampling different nuclear configurations by *ab-initio* MDs is more meaningful than to do it via empirical potential, in fact, if one compares the broad range of structural, dynamical and spectroscopic properties of bulk water obtained from DFT simulations compared to the recently developed potential mb-POL for neat/bulk water, the results are pretty clear: mb-POL consistently performs significantly better than all DFT based simulations. We therefore assume that mb-POL is the best water model on the market for neutral water that should be used, whenever possible, for sampling configurations. mb-POL, however, does not include explicitly electronic degrees of freedom and so if one is interested in understanding phenomena such as charge transfer/polarization, another approach is needed. Another point that hinders the use of mb-POL is our focus on simulations of large-extended interfaces; to do long-scale mb-POL simulations of such an extended air-water interface is not possible at this point. Our strategy was then adapted

to perform long simulations on large systems with classical potentials that still performs better than LS-DFT with respect to the representation of surface tension. Starting from the structures sampled during these simulation the LS-DFT calculations were conducted. For the air-water interface, we validated (see SI) that the charge-transfer mechanism as observed from the asymmetries in the different water defects, is captured both with our empirical simulations and using mb-POL configurations.

2) Related to this point, the authors should comment on how the hydrogen-bond topologies they obtain from their force field calculations compare with previous ab initio molecular dynamics or empirical valence-bond model studies of the air-water or air-oil interface. For example, I believe that in the original study of Kuo and Mundy [Science 303, 658 (2004)] (surprisingly not cited by the authors), it was found that (in the nomenclature of the authors of the present manuscript) 1-in/1-out configurations at the surface layer are important, yet the authors only mention 1-in/0-out and 2-in/1-out topologies. Thus, I think it is important for the authors to address topologies that arise in other studies, compare with theirs, and discuss why they think they have hit on the right topologies. After all, this would seem to be a strong influencing factor in their charge analysis and could affect the results significantly. If possible, comparison with experimentally determined (and possible computationally determined) SFG spectra would also be useful. Are the findings of the authors consistent with what these spectra suggest?

We thank the reviewer for bringing this point up. Indeed, this is a discussion that needs to be added to the manuscript. We now review previous studies on hydrogen bonding patterns at the air-water interface and make an attempt to compare also with what is feasible to compare with experiments. This includes citing the proposed paper by Kuo and Mundy which we missed. We now add a discussion on the similarities and differences between the

topologies observed in that work and ours. Unfortunately, at this point, it is beyond the scope of the current manuscript to link with SFG spectra. This, however, will be the subject of a future study.

A recent experimental work using SFG of the surface of water demonstrated that the number of dangling O-H bonds near the interface, is $\sim 25\%$.³ How one defines the interface that is probed by the SFG experiment is a highly non-trivial task. However, using the definitions proposed by Bonn and coworkers in this recent paper, we computed the number of dangling bonds in the water model we used, TIP4P/2005 - the number of dangling O-H bonds we obtain is $\sim 27\% \pm 2.4\%$ which is consistent with the experiments. Recall, that the the number of water molecules with dangling O-H bonds correspond to the topological defects that are 2in-1out or 1in-1out or 0in-1out (although these are found in very small concentration). Interestingly, Bonn and co-workers³ compare a whole class of empirical water models including mB-POL - the percentage of our dangling O-H bonds is very consistent with the mb-POL predictions. As we have also pointed out from our work, one of the key ingredients of observing the charge asymmetries is the apparent higher concentration of 1in-2out defects versus 2in-1out defects. This feature is not just exclusive to the interface but also in the bulk. In a previous study, we have compared in a very exhaustive study, the concentration of defects in different water models ranging from classical empirical to DFT based and finally, of course mb-POL. Again, the conclusion is that the higher concentration of 1in-2out water molecules compared to 2in-1out is found in *all water models* including mb-POL. This gives us a lot more confidence in our results.

Besides the concentration of these various defects, a factor that plays a very important role is the magnitude of the charge transfer and the fluctuations associated with it. This issue is addressed in detail in response to Reviewer 3's question. We have now added the following

paragraph to the manuscript:

A recent set of new experimental work using SFG of the surface of water demonstrated that the number of dangling O-H bonds near the interface, is $\sim 25\%$. How one defines the interface that is probed by the SFG experiment is non-trivial. However, using the definitions proposed by Bonn and coworkers, we computed the number of dangling bonds in the water model we used namely, TIP4P/2005 - the number of dangling O-H bonds we obtain is $\sim 27\% \pm 2.4\%$ which is consistent with the experiments. Recall, that the number of water molecules with dangling O-H bonds correspond to the topological defects that are 2in-1out or 1in-1out or 0in-1out (although these are found in very small concentration). Interestingly, Bonn and co-workers compare a whole class of empirical water models including mB-POL - the percentage of our dangling O-H bonds is consistent with these results. As we have also pointed out from our work, one of the key ingredients of observing the charge asymmetries is the apparent higher concentration of 1in-2out defects versus 2in-1out defects. This feature is not just exclusive to the interface but also in the bulk. In a previous study,⁴ we have compared exhaustively, the concentration of defects in different water models ranging from classical empirical to DFT based and finally, of course mb-POL. Again, the conclusion is that the higher concentration of 1in-2out water molecules compared to 2in-1out is found in *all water models* including mb-POL.

3) Concerning the charge partitioning scheme, the analysis presented in the main manuscript is based entirely on the DDEC scheme. The SI shows some comparisons between DDEC and NBO for different functionals, but there is really no discussion about why DDEC was chosen and why a comparison between DDEC and NBO should be enough to convince the reader that the scheme was fully vetted. There are numerous charge-partitioning methods available, including old standards such as RESP, and more modern approaches such as ad-

ditive variational Hirshfeld (from the work of Heidar-Zadeh and Ayers), to name a few. In order to avoid the impression that the authors cherry-picked the method (which I'm sure they did not) to support a particular picture, I think what would make a more convincing case would be to present results in the main manuscript that employ several charge partitioning schemes and demonstrate that the overall conclusion about charge transfer among the interfacial layers is not dependent on this choice (even if the results are not entirely the same quantitatively). That is, one would like to see some robustness of this key conclusion rather than having to have faith that DDEC is really producing the right picture.

We thank the reviewer for this comment. Indeed, we have performed an extremely careful and thorough analysis of the sensitivity of the charge transfer we observe to various charge partitioning schemes and the factors that could possibly affect these results. All these results in the original submission, were in the SI. For clarity the reviewer is reminded of all the benchmarks we performed. These include:

- A discussion on the justification of the use of the DDEC charges
- Sampling Protocol: the configurations from which the water coordinates are taken (see discussion earlier)
- Quality of the underlying electronic structure: comparing the results obtained from the GGA functional BLYP to those obtained from hybrid functionals such as B3LYP. In addition some benchmarks were also performed with a an *ab initio* based van-der-Waals functional rVV10 in order to assess the sensitivity of the charge transfer to the inclusion of more accurate van-der-Waals forces. For the small oil-water clusters, we also compared the charge partitioning between the water and oil using wavefunction approaches such as MP2.
- Finally, we also compared the charge partitioning obtained using DDEC charge scheme

to those from an NBO analysis. We also performed some benchmarks using the Iterative Hirshfield (IH) approach to determine how sensitive our charge transfer results are to another charge partitioning scheme as suggested by the reviewer. The reviewer explicitly asked for these results to be moved to the main text where they are now included as shown in Figure 1

Consequently, the following text and Figure have now been added to the main text:

The use of net atomic charges (NEC) is widely used in the area of chemical sciences since they provide a practical and convenient way to partition the electron density distribution into atomic or molecular contributions. The method of choice in this work was the DDEC charge partitioning. In the DDEC method the net atomic charges are assigned as a functional of the electron density hence are not sensitive to the choice of basis set. There are of course many charge schemes to partition the electron density with similar traits such as Bader⁵ and the iterative Hirshfield (IH)⁶ and stockholder (ISA)⁷ approaches. The DDEC method combines the strengths of the ISA and IH methods with the additional trait of giving a more faithful representation of the potential $V(r)$. While some of the criticism around the DDEC methods regards the inelegance of their formulation, their strength relies in an engineering approach that has been shown to lead to sound chemical results.⁸ To assess the reliability of our results with respect to the charge partitioning adopted we compared the results obtained with the DDEC scheme to the ones obtained using two different charge representations: NBO, and IH. These tests were constructed by averaging over 20 and 10 frames for the air/water and oil/water interface respectively, instead of the full 250 and 200 used in the original sets (due to the computational cost). The results depicted in Figure 1 show that, while there are subtle differences between the local behaviour in the different charge representation, the main trends we observe on the charge transfer are consistent across all the three charge schemes.

Figure 1: Comparison of the charge profiles obtained for 20 test frames using IH, NBO and DDEC charge schemes for both water /air and water/oil interfaces (respectively left side and right side panels).

4) Finally, one could levy the criticism that using neutral water is too ideal and not entirely realistic. If we believe previous *ab initio* MD results suggesting a preference of hydronium ions for the interface, then in a real interfacial system, there might be a small population of hydronium ions at the surface, the possibility of which is neglected here. This is not to say that the authors should repeat their study with a hydronium ion, but it would be interesting for them to estimate, based on their results, how the presence of an H_3O^+ ion at the surface (which, admittedly, given their system size, would be a high population) would affect their charge partitioning and hydrogen-bond topologies in the vicinity of the ion and how this might affect their integrated charge densities. Although probably a small perturbation, the results could be sensitive to the presence of this ion.

We thank the reviewer for this question which is an issue that we have also been thinking about. Inserting the hydronium and hydroxide ion into the calculations and examining how it affects the charge transfer mechanisms will be the subject of a forthcoming study. At this point, we can only make speculations about what the presence of a proton would do to the charge at the air-water interface. Assuming that the proton is localized as exists as a local ionic defect, namely on water as an Eigen or on two water molecules as a Zundel, we do not expect it to change the concentration of hydrogen-bond topologies significantly. The details of the charge profile in the presence of the proton will depend a lot on whether it is pinned above the Willard-Chandler interface or if it lies below it. Recent *ab initio* simulations in our group⁴ have shown the possibility of both scenarios: if the proton lies above the instantaneous interface, then the first peak of positive charge would be enhanced; on the other hand, if the proton lies within the second layer, we estimate that one would need a single proton per $\sim 10 \text{ nm}^2$ to neutralize the negative charge caused by the topological defects in that layer. Of course another important ingredient would be where the counter-ion of the proton, for example, the chloride is relative to the interface.

We have now added the following text to the discussion/conclusion section:

Our current models do not account for the possible binding of protons or hydroxide ions to the interface. Inserting the hydronium and hydroxide ion into the calculations and examining how it affects the charge transfer mechanisms will be the subject of a forth coming study. At this point, we can only make speculations about what the presence of a proton would do to the charge at the air-water interface. Assuming that the proton is localized as exists as a local ionic defect, namely on water as an Eigen or on two water molecules as a Zundel, we do not expect it to change the concentration of hydrogen-bond topologies significantly. The details of the charge profile in the presence of the proton will depend a lot on whether it is pinned above the Willard-Chandler interface or if it lies below it. Recent *ab initio* simulations in our group have shown the possibility of both scenarios: if the proton lies above the instantaneous interface, then the first peak of positive charge would be enhanced; on the other hand, if the proton lies within the second layer, we estimate that one would need a single proton per $\sim 10\text{nm}^2$ to neutralize the negative charge caused by the topological defects in that layer. Of course another important ingredient would be where the counter-ion of the proton, for example, the chloride is relative to the interface.

5) There are also a few small technical points the authors should address:

a) Grimme's D2 scheme is not the more accurate of the empirical van-der Waals corrections. It was shown, for example, by Ma and Tuckerman to overestimate the density of bulk water [see J. Chem. Phys. 137, 044506 (2012)]. Why opt for D2 when D3 is superior and, I believe, available in CP2K? Perhaps, for single-point calculations in the water phase, the difference might not be large, but for the oil phase, I would expect a much large effect. Perhaps this could be addressed in the SI.

The DFT calculations performed in this work were done with ONETEP and not CP2K. In any case, for the charge transfer part, neither Grimme D2 nor D3 would cause any differences in the electronic density since these are all post DFT corrections made to the energies and forces. Instead, the rVV10 functional accounts for non-local electronic correlations and therefore van-der-Waals forces more accurately. As shown earlier, our benchmarks with BLYP+D2 and rVV10 yield similar results for the charge transfer.

b) On page 5, the surface tension value of 68.2 needs units.

This has now been fixed: the units were added in the main text (mN/m).

Reviewer # 3 :

This manuscript focuses on how charge redistributes a near an interface, as a result of partial charge transfer (CT) between molecules. This is done for both an oil/water and a vapor/water interface. To successfully address these interface requires large systems and they are able to perform density functional theory (DFT), using about 2000 particles. They use conventional molecular dynamics (MD) to generate configurations, A notable achievement of this work is the DFT calculations on such large systems. They are able to do this using linear scaling DFT. The focus of these calculations is to determine if CT creates a charged interface, as indicated by previous work using potential models (references 24 and 25). This may provide an explanation for electrokinetic properties of air and oil bubbles in water, which appear to have a negative charge. These calculations do find a charge layering- from the top of the interface towards the bulk, first a small positively charged layer, then a larger negatively charged layer, then another positively charged layer, then the neutral bulk. This layering, and the interpretation of that, as coming from hydrogen bond imbalances, is

similar to references 24 and 25, except the present results are an order of magnitude larger, bringing them close to what might be required for the electrokinetic mobilities. They find similar results for the both the vapor/water and oil/water interfaces, except in the oil case, the oil acquires a negative charge. Overall, this is an important, very well carried out and well written paper. I have a few suggestions to make this a stronger paper.

1) On page 10, it is mentioned that the charge transfer mechanism has been proposed in references 7, 24 and 25, but my reading of reference 7 makes no mention of charge transfer..so the 2012 papers (references 24 and 25) were the first to really make this suggestion, with quantitative estimates of charge transfer. Although in 1985 Collins and Washabaugh (Quarterly Review of Biophysics) made the suggestion that charge transfer leads to a negatively charged water surface. (They got some of the details wrong about the direction of charge transfer and the types of hydrogen bonds lost at the surface though). Ref. 7 is solely concerned with rejecting the hydroxide hypothesis.

We thank the reviewer for this positive assessment of our manuscript and the extremely constructive critique given to improve our work. We eliminated reference 7 from that part of the discussion as suggested by the reviewer and updated the manuscript accordingly.

2) I feel that the reasons the surface charges are so much bigger than the previous calculations are not fully explained. From Figure 8 of the supplementary information (I would suggest that figures 8 through 11 are actually tables) it looks like there is a charge transfer of 0.02 or 0.03 e for each hydrogen bond, which is the same as used in references 24 and 25. (A molecule acquires a charge of -0.02 if it donates a hydrogen bond and gets a charge of +0.02 if it accepts a hydrogen bond) . Molecules with even numbers of donated and accepted hydrogen bonds are roughly neutral (as given on Figure 8 of the SI and figure 3 of

the main paper). So why is the charge so much greater than the earlier studies? The authors make the point that even the water molecules with equal number of donated and accepted hydrogen bonds ("2in-2out" or "2 2") have a net charge, but that contribution is not all that great. This is definitely true in the important negatively charged layer (from 0.3 to 1.8 Angstroms above the interface). Here is a largely the difference between negatively charged 1 in - 2out (1 2) and the positively charged 2 in - 1 out (2 1) and there are just more 1 2 than 2 -1, so that layer is negatively charged. Is the case that TIP4P/2005 (the model used to generate the configurations) has much more hydrogen bond imbalances than the SPC/E or charge transfer models used in the previous studies? Is it the use of the Willard-Chandler instantaneous surface, that effectively eliminates surface roughness so that layers are more obvious? (Vacha, et al, J. Phys. Chem. Lett. vol 3, p 107, 2012 found something like this) Or is there some other effect, not captured by charge transfer through hydrogen bonds?

We thank the reviewer for this question. Our original submission did not adequately deal with the factors that make our results different from previous studies. We now address the issues raised by the reviewer in this point. Before we get into the technicalities, it should be stressed that compared to previous studies for the oil-water interface some essential physics was/is missing namely, that there is some non-trivial electronic coupling going on between the two phases which leads to a net charge transfer from the water to the oil.

We first verified from previous literature that the TIP4P/2005 potential does not generate a significant difference in the hydrogen bond topologies with respect to the SPC/E model. Previous work published from one of the authors of the paper⁴ reported (in the SI, Figure S9) that the relative distributions of the coordination defects at the water/air interface obtained for the SPC/E potential. While some differences in population are present they are not significant enough to account for the disparity in charge density found in our study with respect to previous ones.^{9,10} Further comparison between the descriptions of the interface

between these two potentials can be found in Ref.³ Here the authors show in Figure 1, the percentage of the water molecules with free O-H groups at the interface. Again, while a difference (23% for TIP4P/2005 .vs. 26% for SPC/E) is present this will only cause small differences compared to the factors we elucidate in the following.

References^{9,10} both use an upper and lower boundary of 0.02e (-0.02e) as a value to describe the charge transferred along a hydrogen bond. In our charges distributions we found, especially for the second layer (that gives the negative charge oscillation), larger charges than those used in this work. The left most panel of Figure 2 shows the distribution of the charges in the second layer underlying the previous point. In order to compare the different schemes we recomputed the charge contributions coming from individual coordination defects (middle panel of Figure 2) and the total charge in the second layer (right most panel of Figure 2) using the fixed upper boundary of Jungwirth et al.⁷ ($\pm 0.02e$) to we obtain from our simulations.

Figure 2: Gaussian fitting of the 1in-2out (orange), 2in-1out (red) populations in the second layer (left panel). The mean is reported in the upper left part of the panel. Comparison between total charges obtained by our calculations and charges obtained applying the method in Ref⁹ (central panel) The charges obtained using the method in Ref⁹ are labelled with a **b** next to the species name. Net charges obtained for the second layer using ab-initio calculation and the method reported in Ref⁹ (Right panel).

The analysis shown in Figure 2 shows that an increment of the average charge transfer of 50% on each H₂O molecule (and the contributions from the other species that are small but non negligible) leads to a net charge in the layer (Figure 1 rightmost panel) that is much bigger with respect to the one obtained applying a averaged scheme as performed in Ref.⁹

This delicate interplay between charges and populations of difference species has to be carefully considered to get the right net charge for each layer - in our scheme, it is naturally accounted for by solving the quantum mechanical problem of the electrons.

Regarding Ref¹⁰ that CT-FLEX model is based to a charge transfer contribution built on the FLEX scheme.¹¹ In the FLEX scheme, water molecules are kept neutral and in Ref.¹⁰ the author adds a charge transfer contribution that is modulated via *two distance-dependent functions that smoothly go from zero to one*. This contribution is then multiplied for the ab-initio derived valued of 0.02e and redistributed on the basis of equations 6 and 7 (see Ref.¹⁰) between the water molecules atoms. While this nuanced treatment of the charge transfer is more physical, the 0.02e (-0.02e) value set as maximum (minimum) parameter still does not account for the real physics of charge transfer at the interface. In response to the next question by the reviewer, we elucidate that the charge transfer depends in a highly non-trivial manner on several other environmental factors.

3) On page 15, it is mentioned that some features may not be captured by models which use constant charge transfer, not modulated by the environment. Note that the charge transfer models of refs 24 and 25 do have a change transfer that changes with distance, so not constant, and responsive the the local environment, at least in terms of hydrogen bond lengths. It would be interesting to see what other environmental effects might be relevant, other than counting hydrogen bonds. For example, using a tetrahedral order parameter or a Voronoi volume analysis. Does the charge depend on molecule's free volume for instance? Also, note the authors label the 2 in 2 out molecules as "tetrahedral" but having that hydrogen bond structure does not necessary mean a tetrahedral local geometry, so better to avoid that term unless an analysis does show they are tetrahedral.

The reviewer asks essentially what physics is missing from the charge transfer models re-

ported in previous works.^{9,10} It is true however that Ref.¹⁰ model has a parametrisation that is distance dependent but for the case of Ref.,⁹ the charge transfer only depends by the presence of an hydrogen bond asymmetry. We have thus mended the manuscript in the following way editing the the original sentence: These types of features would not be captured by models using a constant charge transfer value^{9,10} that is not modulated by the local environment. to now read as: These types of features would not be captured by models using a constant⁹ or parametrised charge transfer¹⁰ schemes that only partially respond to the local environment losing important collective polarization effects.

In order to explore if other environmental parameters are relevant in determining the charge fluctuations on the H₂O molecules, we correlated several geometrical descriptors of the environment around the water molecules, to the magnitude of the net charge observed on each water molecule and their distance with respect from the Williard-Chandler instantaneous surface. Some of these correlation analysis are shown below for the standard 2in-2out waters and the important defects : 0in-1out, 1in-0out, 1in-1out, 1in-2out and 2out-1in (where the population is big enough to obtain a proper correlation). Our manuscript is rather long already so we will present a summarised version of the pictures/notes to follow.

Figure 3: Panel A: Angular features for a tetrahedral coordinated water as reported in Ref. ¹² The A tag is referred to the H₂O molecules accepting an hydrogen bond by the central water. The D tag is referred to the H₂O molecules donating an hydrogen bond to the central water. The subscript to each tag refers to the distance from the central water (i.e. A₁ is the nearest water to the central one accepting an OH bond from it). Panel B: Illustration of the proton transfer coordinates (PTC) in elliptical coordinates reproduced from Ref. ¹³

Figures 4, 5, 6, 7 show the correlation between different geometrical parameters (see Figure 3), the total charge on each H₂O molecule (WATC) and its distance from the Willard-Chandler distance (WCD). The geometrical descriptors chosen have been previously reported in. ^{12,13} These descriptors were chosen since tetrahedrality ^{14,15} and Local Structure Index ¹⁶ struggled to capture the geometry of the interfacial water HB network. In fact due to the strained configurations present at the water/air boundary the significance hold by these measurements become unclear. The proton transfer coordinate (PTC) in elliptical coordinates (τ) shows a very strong negative correlation with respect to the central H₂O WATC for those water molecules accepting an HB from it (i.e. A _{τ 1}, A _{τ 2}). A strong positive correlation is instead observed for waters molecules donating an HB to the central H₂O (i.e. D _{τ 1}, D _{τ 2}). These results shows that the more the proton is shared (equivalent to smaller τ) between the molecules accepting an HB and the central H₂O the higher is the charge transferred to it making WATC more negative (this behaviour confirms the same trend observed in bulk

Figure 4: Correlation coefficients (Pearson) calculated between the central water total charge (WATC) and the features shown in 3 for the most relevant water coordination defects in the first layer. The shorthanded labels for the PTC are: $D_{\tau}x$ and $D_{\sigma}x$ that indicate the quantities showed in panel B for D1 and D2 respectively. The same notation is applied for A1 and A2 (i.e. $A_{\tau}x$, $A_{\sigma}x$). WCD is the distance of the water molecule with respect to the Willard Chandler interface. For the angular features AA1 represents the $\widehat{A1CA2}$ angle, DD1 the $\widehat{D1CD2}$ angle, AD1 the $\widehat{A1CD1}$ angle, AD2 the $\widehat{A1CD2}$ angle, AD3 the $\widehat{A2CD1}$ angle and AD4 is the $\widehat{A1CD1}$ angle.

water as showed in Ref.¹³). In the case of $D_{\tau}1$ and $D_{\tau}2$ the same trend stands however the charge is transferred from the central water to the other H_2O molecules making it more positive hence we observe a positive correlation.

Passing to the angular descriptors we observe the positive correlation between WATC and the angle defined by the central water molecule, $A_{\tau}1$ and $A_{\tau}2$. Conversely, negative correlation is observed for the angle between the central H_2O , $D_{\tau}1$, $D_{\tau}2$. These relations seems to suggest that the bigger is the angle formed by the central water molecule and the H_2O s accepting an HB from it the more positive is WATC. This effect could be ascribed to a more effective alignment of the bond dipoles that results in a bigger charge transfer from the

Figure 5: Correlation coefficients (Paerson) calculated between the central water total charge (WATC) and the features shown in 3 for the most relevant water coordination defects in the 2nd layer. The shorthanded labels for the PTC are: $D_{\tau}x$ and $D_{\sigma}x$ that indicate the quantities showed in panel B for D1 and D2 respectively. The same notation is applied for A1 and A2 (i.e. $A_{\tau}x$, $A_{\sigma}x$). WCD is the distance of the water molecule with respect to the Willard Chandler interface. For the angular features AA1 represents the $\widehat{A1CA2}$ angle, DD1 the $\widehat{D1CD2}$ angle, AD1 the $\widehat{A1CD1}$ angle, AD2 the $\widehat{A1CD2}$ angle, AD3 the $\widehat{A2CD1}$ angle and AD4 is the $\widehat{A1CD1}$ angle.

Figure 6: Correlation coefficients (Paerson) calculated between the central water total charge (WATC) and the features shown in 3 for the most relevant water coordination defects in the 3rd layer. The shorthanded labels for the PTC are: $D_{\tau}x$ and $D_{\sigma}x$ that indicate the quantities showed in panel B for D1 and D2 respectively. The same notation is applied for A1 and A2 (i.e. $A_{\tau}x$, $A_{\sigma}x$). WCD is the distance of the water molecule with respect to the Willard Chandler interface. For the angular features AA1 represents the $A1\widehat{CA}2$ angle, DD1 the $D1\widehat{CD}2$ angle, AD1 the $A1\widehat{CD}1$ angle, AD2 the $A1\widehat{CD}2$ angle, AD3 the $A2\widehat{CD}1$ angle and AD4 is the $A1\widehat{CD}1$ angle.

central H_2O to its coordinated counterparts. Similar reasoning but with the opposite sign can be applied for the molecules donating an HB to the central H_2O . A trend regarding the other angular parameters is generally difficult to find.

At last we consider the correlation of WCD with WATC and the geometric descriptors. Figures 4, 5, 6 and 7 don't give a clear picture to clearly draw a direct correlation between WATC, the angular descriptors and WCD for each water molecule.

On the basis of the observations reported above, we amended the manuscript adding the following text under the section "Charge Partition Schemes and Correlations":

Figure 7: Correlation coefficients (Paerson) calculated between the central water total charge (WATC) and the features shown in 3 for the most relevant water coordination defects in the 3rd layer. The shorthanded labels for the PTC are: $D_{\tau}x$ and $D_{\sigma}x$ that indicate the quantities showed in panel B for D1 and D2 respectively. The same notation is applied for A1 and A2 (i.e. $A_{\tau}x$, $A_{\sigma}x$). WCD is the distance of the water molecule with respect to the Willard Chandler interface. For the angular features AA1 represents the $\widehat{A1CA2}$ angle, DD1 the $\widehat{D1CD2}$ angle, AD1 the $\widehat{A1CD1}$ angle, AD2 the $\widehat{A1CD2}$ angle, AD3 the $\widehat{A2CD1}$ angle and AD4 is the $\widehat{A1CD1}$ angle.

In order to understand how the charge fluctuations are modulated by the local environment, we determined the correlation between the total charge on each H_2O molecule (WATC) and various geometrical descriptors which are visually depicted in Figure 8 A-B). Figure 8 C) shows the Pearson correlation coefficient between the different geometrical parameters and WATC for the most populated water molecules in the different water layers relative to

Figure 8: Panel A: Angular coordinates for a tetrahedral coordinated water as reported in Ref.¹² The A tag is referred to the H₂O molecules accepting a hydrogen bond from the central water. The D tag is referred to the H₂O molecules donating an hydrogen bond to the central water. The subscript for each tag refers to the distance from the central water (i.e. A₁ is the nearest water to the central one accepting an OH bond from it). Panel B: Illustration of the proton transfer coordinates (PTC) in elliptical coordinates reproduced from Ref.¹³ Panel C: Most relevant correlation coefficients (Pearson) calculated between the charge of the central water molecule (WATC) and the features shown in Panel A and B. Only the results for the 2in-1out H₂O in the 1st layer (1st row), 1in-2out in the 2nd layer (2nd row), 2in-2out in the 3rd and 4th layers (3rd and 4th row respectively) are reported for brevity. The full correlation matrices for each species in every layer are reported in the SI. The shorthanded labels for the PTC are: $D_{\tau x}$ and $D_{\sigma x}$ that indicate the quantities shown in panel B for D1 and D2 respectively. The same notation is applied for A1 and A2 (i.e. $A_{\tau x}$, $A_{\sigma x}$). WCD is the distance of the water molecule with respect to the Willard Chandler interface. For the angular features, AA1 represents the $\widehat{A1CA2}$ angle, DD1 the $\widehat{D1CD2}$ angle, AD1 the $\widehat{A1CD1}$ angle, AD2 the $\widehat{A1CD2}$ angle, AD3 the $\widehat{A2CD1}$ angle and AD4 is the $\widehat{A1CD1}$ angle.

the interface. The four rows in the table correspond to the four regions relative to the WCD as shown in Figure 3. For a more exhaustive analysis of the correlation matrices in all the layers, the reader is referred to the SI. The proton transfer coordinate (PTC)¹³ (τ) shows a

strong negative correlation with respect to the central H₂O WATC for those water molecules accepting a HB from it (i.e. $A_{\tau 1}$, $A_{\tau 2}$). A strong positive correlation is instead observed for waters molecules donating a HB to the central H₂O (i.e. $D_{\tau 1}$, $D_{\tau 2}$). These results show that the more the proton is shared, namely a smaller τ between the molecules accepting a HB and the central H₂O, the higher is the charge transferred to it making the central water more negatively charged. For the case of $D_{\tau 1}$ and $D_{\tau 2}$, the proton transfer coordinate plays a similar role but now the charge is transferred from the central water to the other H₂O molecules making it more positive. For this reason, we observe a positive correlation in this case. Although the charge transfer is much less correlated with the angular descriptors, the effects are not insignificant. We observe a positive correlation between WATC and the angle defined by the central water molecule, $A_{\tau 1}$ and $A_{\tau 2}$. Conversely, a negative correlation is observed for the angle between the central H₂O, $D_{\tau 1}$, $D_{\tau 2}$. These relations seems to suggest that the bigger is the angle formed by the central water molecule and the H₂O's accepting a HB from it, the more positive is WATC. This effect could be ascribed to a more effective alignment of the bond dipoles that results in a larger charge transfer from the central H₂O to its hydrogen bond partners. This is also reflected in the correlation between the charge of the central water and AA1 angle described in the caption of Figure 8. Clearly, the fact that the correlations we report along these geometrical parameters represent only a subset of the reaction coordinates involved in modulating the charge transfer and warrants further investigation.

4) On the bottom of page 14, they refer to Figures 7 through 11 of the SI, but they must mean figures 8 through 11 (and they should really be tables, as suggested above).

We thank the reviewer for spotting this imprecision and updated the manuscript accordingly. The Figures figures 8 through 11 have been renamed Tables and the main text has been amended.

With some further clarification about why the charges are so much bigger than previous estimates this would be a stronger paper. Note that the magnitude is important if it is the explanation for the electrokinetic results. Otherwise, this is an interesting, important and well-done paper

References

- (1) Ojha, D.; Karhan, K.; Khne, T. D. On the Hydrogen Bond Strength and Vibrational Spectroscopy of Liquid Water. *Scientific Reports* **2018**, *8*, 16888.
- (2) Ito, H.; Hasegawa, T.; Tanimura, Y. Effects of Intermolecular Charge Transfer in Liquid Water on Raman Spectra. *The Journal of Physical Chemistry Letters* **2016**, *7*, 4147–4151, PMID: 27689824.
- (3) Sun, S.; Tang, F.; Imoto, S.; Moberg, D. R.; Ohto, T.; Paesani, F.; Bonn, M.; Backus, E. H. G.; Nagata, Y. Orientational Distribution of Free O-H Groups of Interfacial Water is Exponential. *Phys. Rev. Lett.* **2018**, *121*, 246101.
- (4) Giberti, F.; Hassanali, A. A. The excess proton at the air-water interface: The role of instantaneous liquid interfaces. *J. Chem. Phys.* **2017**, *146*, 244703.

- (5) Tang, W.; Sanville, E.; Henkelman, G. A grid-based Bader analysis algorithm without lattice bias. *Journal of Physics: Condensed Matter* **2009**, *21*, 084204.
- (6) Bultinck, P.; Van Alsenoy, C.; Ayers, P. W.; Carbo-Dorca, R. Critical analysis and extension of the Hirshfeld atoms in molecules. *The Journal of Chemical Physics* **2007**, *126*, 144111.
- (7) Lillestolen, T. C.; Wheatley, R. J. Redefining the atom: atomic charge densities produced by an iterative stockholder approach. *Chem. Commun.* **2008**, 5909–5911.
- (8) Heidar-Zadeh, F.; Ayers, P. W.; Verstraelen, T.; Vinogradov, I.; Vhringer-Martinez, E.; Bultinck, P. Information-Theoretic Approaches to Atoms-in-Molecules: Hirshfeld Family of Partitioning Schemes. *The Journal of Physical Chemistry A* **2018**, *122*, 4219–4245, PMID: 29148815.
- (9) Vcha, R.; Marsalek, O.; Willard, A. P.; Bonthuis, D. J.; Netz, R. R.; Jungwirth, P. Charge transfer between water molecules as the possible origin of the observed charging at the surface of pure water. *J Phys. Chem. Lett.* **2012**, *3*, 107–111.
- (10) Wick, C. D.; Lee, A. J.; Rick, S. W. How intermolecular charge transfer influences the air-water interface. *J. Chem. Phys.* **2012**, *137*, 154701.
- (11) Wick, C. D. Hydronium Behavior at the AirWater Interface with a Polarizable Multi-state Empirical Valence Bond Model. *The Journal of Physical Chemistry C* **2012**, *116*, 4026–4038.
- (12) Shin, S.; Willard, A. P. Three-Body Hydrogen Bond Defects Contribute Significantly to the Dielectric Properties of the Liquid WaterVapor Interface. *The Journal of Physical Chemistry Letters* **2018**, *9*, 1649–1654, PMID: 29528654.

- (13) Schran, C.; Marsalek, O.; Markland, T. E. Unravelling the influence of quantum proton delocalization on electronic charge transfer through the hydrogen bond. *Chemical Physics Letters* **2017**, *678*, 289 – 295.
- (14) Chau, P.-L.; Hardwick, A. A new order parameter for tetrahedral configurations. *Mol. Phys.* **1998**, *93*, 511–518.
- (15) Errington, J. R.; Debenedetti, P. G. Relationship between structural order and the anomalies of liquid water. *Nature* **2001**, *409*, 318–321.
- (16) Santra, B.; Jr., R. A. D.; Martelli, F.; Car, R. Local structure analysis in ab initio liquid water. *Molecular Physics* **2015**, *113*, 2829–2841.

REVIEWERS' COMMENTS:

Reviewer #2 (Remarks to the Author):

I have read the replies by the authors and the revised manuscript. The new version is really a great read and will make an excellent contribution to this ongoing debate. I recommend publication of the manuscript as it now stands.

Reviewer #3 (Remarks to the Author):

Overall, I feel the authors responded well to the suggestions of the reviewers and this is an interesting and important paper, worthy of publication in Nature Communications. I have a few revisions to suggest.

1. On page 3, the statement "At the surface of water, the charge transfer leads to a triple layer of charge with negative surface charge a couple of Angstroms from the surface" needs references. Also the next sentence about a similar effect at the oil/water interface needs references.
2. The simulations using the mb-POL model could be better described. In the main manuscript, these studies seem like an afterthought. A short description of the model could be helpful. I'm assuming it has many body terms and is polarizable, but how does it deal with hydrogens, for example. Is it a rigid model? This might be relevant in discussing the charge transfer effects, since the (rigid) TIP4P/2005 model used for most of simulations.
3. On page 23, the charge transfer as a function of the tau coordinate is discussed. The statement is made that "the more a proton is shared....the higher is the charge transferred to it." Just to be clear, this means more charge is transferred to the water molecule donating the hydrogen bond? Also, since the water models are rigid, (so dOH in Figure 9 is a constant) a more shared proton just means a shorter hydrogen bond length, so does this just mean as the hydrogen bond length gets shorter there is more charge transfer? Or there something else going on?
4. In the last paragraph, on page 26, speculations about hydroxide and hydronium ions, and counter-ions are made. In connection to this, previous work using a combination of empirical valence bond, quantum calculations, and charge transfer models have looked at the effects of charge transfer near the water/vapor interface with a hydroxide or hydronium ion present (J. Chem. Phys. Vol 143, 044702, 2015). The charge transfer effects of other ions and counter-ions near the surface have also been studied (J. Chem. Phys. vol. 140, 184703, 2014). These studies might help the speculations made here, and be useful for the further studies. Those papers present results on how charge transfer differs whether the ions are above or below the instantaneous surface.
5. On page 14 of the Supplementary Information, charge transfer models using a fixed amount of charge transfer are referenced by the Jungwirth 2012 paper (ref. 8). It should be noted that this model originated in papers published in 2011 (J. Chem. Phys. vol. 134, 184507, 2011 and JACS vol. 133, 10204, 2011).
6. The differences between the simple, hydrogen-bond based charge transfer (CT) models and the quantum calculations of this work are discussed in the Supplementary Information on page 14 and Figure 8. Their ab initio calculations give an average charge for a hydrogen bond imbalance equal to about 0.03 e, bigger than the CT model value of 0.02 e. The total charge of the second layer under the surface is compared using the CT model and the ab initio results. Given that the average charge transfer amount is only 50% bigger, it is still a little unclear to me why the net charge of the layer is about 7 times larger (about -0.35 e vs -0.05 e). It seems like the difference should be 50% at best, since the 1in-2out positive charges and the 2in -1out negative charges, both bigger for the ab initio results, should partially cancel. Which means that there is more leading to charge transfer than hydrogen bond imbalance. So I don't really agree with the statement that "an increment of the average charge transfer of 50%...leads to a net charge in the layer that is much bigger with respect to the one obtained using the average scheme." I would suggest they remove or modify that sentence. Understanding those other factors is the point of Figures 9 through 12, but it doesn't seem obvious what the main factor is that leads to the order of magnitude bigger

surface charges.

7. In this part of the Supplementary Information, much of it seems a verbatim repeat of material in the main manuscript (page 16 for example). Maybe that wasn't intentional. Also on page 15, the authors mention a response to reviewers, that is probably not intentional either.

Charge Transfer as a Ubiquitous Mechanism in Determining the Negative Charge at Hydrophobic Interfaces

Emiliano Poli, Kwang H. Jong, Ali Hassanali

mail: epoli@ictp.it, ahassanali@ictp.it

Condensed Matter Statistical Physics Department, The Abdus Salam International Center for Theoretical Physics
Strada Costiera, Trieste

December 30, 2019

1 Answer to Reviewer 2

We thank the Reviewer 2 for the positive judgment of our work.

2 Answer to Reviewer 3

Overall, I feel the authors responded well to the suggestions of the reviewers and this is an interesting and important paper, worthy of publication in Nature Communications. I have a few revisions to suggest.

We thank the reviewer for the positive judgment on our work.

1. On page 3, the statement “At the surface of water, the charge transfer leads to a triple layer of charge with negative surface charge a couple of Angstroms from the surface” needs references. Also the next sentence about a similar effect at the oil/water interface needs references.

We thank the reviewer for the suggestion however we would like to stress that these statements refer to our own results shown in Figure 2 panel a and b. We added a reference to the picture in the main text to underline this point. The presence of the triple layer is also consistent with previous studies and we now add some references associated with this sentence.

2. The simulations using the mb-POL model could be better described. In the main manuscript, these studies seem like an afterthought. A short description of the model could be helpful. I’m assuming it has many body terms and is polarizable, but how does it deal with hydrogens, for example. Is it a rigid model? This might be relevant in discussing the charge transfer effects, since the (rigid) TIP4P/2005 model used for most of simulations.

We thank the reviewer for pointing out this unclear part of our manuscript. We want to stress that we did describe the mb-POL simulations in the Supporting Information of the manuscript. These were only used as verification against our results given the smaller dimension of the system we were able to simulate. Such limitations did not permit an analysis as extensive as the one for the main systems simulated with TIP4P/2005 potential. It is also worth noting that it is currently also not possible to perform mb-POL simulations with hydrocarbons such as dodecane. For this reason, we decided that the relevance with respect to most of the results in the main manuscript was modest. In order to address this point raised by the reviewer we inserted an additional sentence in the Methods section:

Besides the quality of the electronic structure, we also examined the sensitivity of the charge transfer to sampling configurations of the air-water interface sampled from mb-POL. The mb-POL potential reproduces many structural and dynamical properties of water across the phase diagram and while we could not simulate a system as big as the one with TIP4P/2005, this test provide a useful comparison with

respect to our results. More details about these benchmark simulations are expanded upon in the SI.

and expanded the description of the mb-POL model in the Supporting Information:

mb-POL is a *first principles* based water potential with flexible monomers for molecular simulations of water systems from the gas to condensed phase. MB-pol explicitly treats the one-body term and the short-ranged two- and three-body interaction terms. mb-POL accurately describes the properties of gas-phase clusters, such as the dimer vibration-rotation tunneling spectrum, the second and third virial coefficients of water as well as cluster structures and energies[]. In addition, mb-POL gives a highly accurate description of the liquid phase of water at ambient conditions in comparison with experiment for several structural, thermodynamic, dynamical and spectroscopic properties[].

3. On page 23, the charge transfer as a function of the tau coordinate is discussed. The statement is made that “the more a proton is shared...the higher is the charge transferred to it.” Just to be clear, this means more charge is transferred to the water molecule donating the hydrogen bond? Also, since the water models are rigid, (so dOH in Figure 9 is a constant) a more shared proton just means a shorter hydrogen bond length, so does this just mean as the hydrogen bond length gets shorter there is more charge transfer? Or there something else going on?

We thank the reviewer for this observation. We would like to confirm that indeed the reviewer is correct. Since the water model we used is rigid, a more shared proton is equivalent to having a shorter hydrogen bond length. The reviewer is also correct in interpreting that the electronic charge is transferred from the H₂O accepting the Hydrogen Bond to the one donating it.

4. In the last paragraph, on page 26, speculations about hydroxide and hydronium ions, and counterions are made. In connection to this, previous work using a combination of empirical valence bond, quantum calculations, and charge transfer models have looked at the effects of charge transfer near the water/vapor interface with a hydroxide or hydronium ion present (J. Chem. Phys. Vol 143, 044702, 2015). The charge transfer effects of other ions and counter-ions near the surface have also been studied (J. Chem. Phys. vol. 140, 184703, 2014). These studies might help the speculations made here, and be useful for the further studies. Those papers present results on how charge transfer differs whether the ions are above or below the instantaneous surface.

We thank the reviewer for the helpful suggestions. After reading the two papers indicated we modified the text as follows:

Assuming that the proton exists as a local ionic defect, namely on a single water molecule as an Eigen or on two water molecules as a Zundel, we do not expect it to change the concentration of hydrogen-bond topologies significantly. However, the presence of the ions will likely also change the extent of charge transfer between the water molecules at the interface as observed in previous studies[2, 1]. The details of the charge profile in the presence of the proton will also depend a lot on whether it is pinned above the Willard-Chandler interface or if it lies below it. If the proton lies within the second layer, we estimate that one would need a single proton per ~ 10 nm² to neutralize the negative charge caused by the topological defects in that layer. Of course another important ingredient would be the role of counterions. Previous studies have shown that while charge transfer does not affect the binding affinity of ions to the surface of water, ions can have long-range effects on surface charge[2]. More specific answers to these questions will be the subject of a forthcoming study.

5. On page 14 of the Supplementary Information, charge transfer models using a fixed amount of charge transfer are referenced by the Jungwirth 2012 paper (ref. 8). It should be noted that this model originated in papers published in 2011 (J. Chem. Phys. vol. 134, 184507, 2011 and JACS vol. 133, 10204, 2011).

We thank the reviewer for highlighting this imprecision. The reference was corrected and modified.

6. The differences between the simple, hydrogen-bond based charge transfer (CT) models and the quantum calculations of this work are discussed in the Supplementary Information on page 14 and Figure 8. Their ab initio calculations give an average charge for a hydrogen bond imbalance equal to about 0.03

e, bigger than the CT model value of 0.02 e. The total charge of the second layer under the surface is compared using the CT model and the ab initio results. Given that the average charge transfer amount is only 50% bigger, it is still a little unclear to me why the net charge of the layer is about 7 times larger (about -0.35 e vs -0.05 e). It seems like the difference should be 50% at best, since the 1in-2out positive charges and the 2in-1out negative charges, both bigger for the ab initio results, should partially cancel. Which means that there is more leading to charge transfer than hydrogen bond imbalance. So I don't really agree with the statement that "an increment of the average charge transfer of 50%...leads to a net charge in the layer that is much bigger with respect to the one obtained using the average scheme." I would suggest they remove or modify that sentence. Understanding those other factors is the point of Figures 9 through 12, but it doesn't seem obvious what the main factor is that leads to the order of magnitude bigger surface charges.

We thank the reviewer for taking a careful look at the data. We agree with the reviewer when he states that the effect and differences in charge that we observe are due to more than just an increment in the average of the 1in-2out charges and the 2in-1out negative charges. The differences arise from a combination of various factors: i) firstly, the relative concentration of 1in-2out vs 2in-1out defects - contrary to common intuition, there is a larger concentration of 1in-2out vs 2in-1out water molecules (a property that ends up being present in the bulk as well and found in all water models), ii) differences in the shape of the distributions of the different species as reported by the skewness and kurtosis of the charge distributions and finally iii) the larger average charge transfer. It should be stressed that in the manuscript, we do not specify that this as the sole source of the difference in charge but instead. We now add the following text to the first paragraph of the conclusion to clarify these points:

The large surface charges we observe compared to previous studies[], arises from a combination of several different factors such as the higher concentration of 1in-2out versus 2in-1out water molecules, the larger average charge associated with each water species and finally, the asymmetries associated with the fluctuations in the charge of these water molecules. It is also clear from the analysis of the charge and correlations with geometrical parameters that the charge transfer is a rather complex process involving the coupling between several different degrees of freedom.

7. In this part of the Supplementary Information, much of it seems a verbatim repeat of material in the main manuscript (page 16 for example). Maybe that wasn't intentional. Also on page 15, the authors mention a response to reviewers, that is probably not intentional either.

We thank the reviewer for underlying these similarities and imprecision. Regarding the repetitions we tried to ameliorate them however some part are bound to be similar since in the main text we reported a summary of the data shown in the Supporting Information for which some of the wording was used as a conclusion.

References

- [1] Marielle Soniat, Revati Kumar, and Steven W. Rick. Hydrated proton and hydroxide charge transfer at the liquid/vapor interface of water. *The Journal of Chemical Physics*, 143(4):044702, 2015.
- [2] Marielle Soniat and Steven W. Rick. Charge transfer effects of ions at the liquid water/vapor interface. *The Journal of Chemical Physics*, 140(18):184703, 2014.